# Wisdom of Committees: An Overlooked Approach To Faster and More Accurate Models

**Xiaofang Wang[1,2]*  Dan Kondratyuk[1]†  Eric Christiansen[1]**
**Kris M. Kitani[2]  Yair Alon (prev. Movshovitz-Attias)[1]  Elad Eban[1]**
[1]Google Research  [2]Carnegie Mellon University
{xiaofan2,kkitani}@cs.cmu.edu
{dankondratyuk,ericmc,yairmov,elade}@google.com

## Abstract

Committee-based models (ensembles or cascades) construct models by combining existing pre-trained ones. While ensembles and cascades are well-known techniques that were proposed before deep learning, they are not considered a core building block of deep model architectures and are rarely compared to in recent literature on developing efficient models. In this work, we go back to basics and conduct a comprehensive analysis of the efficiency of committee-based models. We find that even the most simplistic method for building committees from existing, independently pre-trained models can match or exceed the accuracy of state-of-the-art models while being drastically more efficient. These simple committee-based models also outperform sophisticated neural architecture search methods (e.g., BigNAS). These findings hold true for several tasks, including image classification, video classification, and semantic segmentation, and various architecture families, such as ViT, EfficientNet, ResNet, MobileNetV2, and X3D. Our results show that an EfficientNet cascade can achieve a 5.4x speedup over B7 and a ViT cascade can achieve a 2.3x speedup over ViT-L-384 while being equally accurate.

## 1 Introduction

Optimizing the efficiency of neural networks is important for real-world applications as they can only use limited computational resources and often have requirements on response time. There has been considerable work in this direction (Howard et al., 2017; Zhang et al., 2018; Tan & Le, 2019), but they mostly focus on designing novel network architectures that can achieve a favorable speed-accuracy trade-off. Here, we do not present any novel method or architecture design. Instead, we focus on analyzing the accuracy and efficiency of a simple paradigm: committee-based models. We use the term "committee" to refer to model ensembles or cascades, which indicates that they are built using *multiple* independent models.

Committee-based models have been extensively studied and used before deep learning (Breiman, 1996; Schapire, 1990; Freund & Schapire, 1997; Viola & Jones, 2001). However, when com-

Figure 1: Committee-based models achieve a higher accuracy than single models on ImageNet while using fewer FLOPs. For example, although Inception-v4 ('Incep-v4') outperforms all single ResNet models, a ResNet cascade can still outperform Incep-v4 with fewer FLOPs.

---

*Work done during an internship at Google.
†Work done as part of the Google AI Residency Program.

paring the efficiency of deep models, committee-based models are rarely considered in recent work (Howard et al., 2017; Zhang et al., 2018; Tan & Le, 2019). There still lacks a systematic understanding of their efficiency in comparison with single models – models that only use one network. Such an understanding is informative for both researchers to push the frontier of efficient models and practitioners to select model designs in real-world applications.

To fill this knowledge gap, we conduct a comprehensive analysis of the efficiency of committee-based models. To highlight the practical benefit of committee-based models, we intentionally choose the simplest possible method, which directly uses off-the-shelf, independently pre-trained models to build ensembles or cascades. We ensemble multiple pre-trained models via a simple average over their predictions (Sec. 3). For cascades, we sequentially apply each model and use a simple heuristic (*e.g.*, maximum probability in the prediction) to determine when to exit from the cascade (Sec. 4).

We show that even this method already outperforms state-of-the-art architectures found by costly neural architecture search (NAS) methods. Note that this method works with off-the-shelf models and does not use specialized techniques. For example, it differs from Boosting (Schapire, 1990) where each new model is conditioned on previous ones, and does not require the weight generation mechanism in previous efficient ensemble methods (Wen et al., 2020). This method does not require the training of an early exit policy (Bolukbasi et al., 2017; Guan et al., 2018) or the specially designed multi-scale architecture (Huang et al., 2018) in previous work on building cascades.

To be clear, the contribution of this paper is not in the invention of model ensembles and cascades, as they have been known for decades, and is not in a new proposed method to build them. Instead, it is in the thorough evaluation and comparison of committee-based models with commonly used model architectures. Our analysis shows that committee-based models provide a simple complementary paradigm to achieve superior efficiency without tuning the architecture. One can often improve accuracy while reducing inference and training cost by building committees out of existing networks.

Our findings generalize to a wide variety of tasks, including image classification, video classification, and semantic segmentation, and hold true for various architecture families: ViT (Dosovitskiy et al., 2021), EfficientNet (Tan & Le, 2019), ResNet (He et al., 2016), MobileNetV2 (Sandler et al., 2018), and X3D (Feichtenhofer, 2020). We summarize our findings as follows:

- Ensembles are more cost-effective than a single model in the large computation regime (Sec. 3). For example, an ensemble of two separately trained EfficientNet-B5 models matches B7 accuracy, a state-of-the-art ImageNet model, while having almost 50% less FLOPs (20.5B vs. 37B).

- Cascades outperform single models in *all* computation regimes (Sec. 4&5). Our cascade matches B7 accuracy while using on average 5.4x fewer FLOPs. Cascades can also achieve a 2.3x speedup over ViT-L-384, a Transformer architecture, while matching its accuracy on ImageNet.

- We further show that (1) the efficiency of cascades is evident in both FLOPs and on-device latency and throughput (Sec. 5.1); (2) cascades can provide a guarantee on worst-case FLOPs (Sec. 5.2); (3) one can build self-cascades using a single model with multiple inference resolutions to achieve a significant speedup (Sec. 6).

- Committee-based models are applicable beyond image classification (Sec. 7) and outperform single models on the task of video classification and semantic segmentation. Our cascade outperforms X3D-XL by 1.2% on Kinetics-600 (Carreira et al., 2018) while using fewer FLOPs.

## 2   RELATED WORK

**Efficient Neural Networks.** There has been significant progress in designing efficient neural networks. In early work, most efficient networks, such as MobileNet (Howard et al., 2017; Sandler et al., 2018) and ShuffleNet (Howard et al., 2019), were manually designed. Recent work started to use neural architectures search (NAS) to automatically learn efficient network designs (Zoph et al., 2018; Cao et al., 2019; Tan et al., 2019; Tan & Le, 2019; Chaudhuri et al., 2020). They mostly fcous on improving the efficiency of single models by designing better architectures, while we explore committee-based models without tuning the architecture.

**Ensembles.** Ensemble learning has been well studied in machine learning and there have been many seminal works, such as Bagging (Breiman, 1996), Boosting (Schapire, 1990), and AdaBoost (Freund & Schapire, 1997). Ensembles of neural networks have been used for many tasks, such as image

classification (Szegedy et al., 2015; Huang et al., 2017a), machine translation (Wen et al., 2020), active learning (Beluch et al., 2018), and out-of-distribution robustness (Lakshminarayanan et al., 2017; Fort et al., 2019; Wenzel et al., 2020). But the efficiency of model ensembles has rarely been systematically investigated. Recent work indicated that ensembles can be more efficient than single models for image classification (Kondratyuk et al., 2020; Lobacheva et al., 2020). Our work further substantiates this claim through the analysis of modern architectures on large-scale benchmarks.

**Cascades.** A large family of works have explored using cascades to speed up certain tasks. For example, the seminal work from Viola & Jones (2001) built a cascade of increasingly complex classifiers to speed up face detection. Cascades have also been explored in the context of deep neural networks. Bolukbasi et al. (2017) reduced the average test-time cost by learning a policy to allow easy examples to early exit from a network. A similar idea was also explored by Guan et al. (2018). Huang et al. (2018) proposed a specially designed architecture Multi-Scale DenseNet to better incorporate early exits into neural networks. Given a pool of models, Streeter (2018) presented an approximation algorithm to produce a cascade that can preserve accuracy while reducing FLOPs and demonstrated improvement over state-of-the-art NAS-based models on ImageNet. Different from previous work that primarily focuses on developing new methods to build cascades, we show that even the most straightforward method can already provide a significant speedup without training an early exit policy (Bolukbasi et al., 2017; Guan et al., 2018) or designing a specialized multi-scale architecture (Huang et al., 2018).

**Dynamic Neural Networks.** Dynamic neural networks allocate computational resources based on the input example, *i.e.*, spending more computation on hard examples and less on easy ones (Han et al., 2021). For example, Shazeer et al. (2017) trained a gating network to determine what parts in a high-capacity model should be used for each example. Recent work (Wu et al., 2018; Veit & Belongie, 2018; Wang et al., 2018) explored learning a policy to dynamically select layers or blocks to execute in ResNet based on the input image. Our analysis shows that cascades of pre-trained models are actually a strong baseline for dynamic neural networks.

## 3 ENSEMBLES ARE ACCURATE, EFFICIENT, AND FAST TO TRAIN

Model ensembles are useful for improving accuracy, but the usage of multiple models also introduces extra computational cost. When the total computation is fixed, which one will give a higher accuracy: single models or ensembles? The answer is important for real-world applications but this question has rarely been systematically studied on modern architectures and large-scale benchmarks.

We investigate this question on ImageNet (Russakovsky et al., 2015) with three architecture families: EfficientNet (Tan & Le, 2019), ResNet (He et al., 2016), and MobileNetV2 (Sandler et al., 2018). Each architecture family contains a series of networks with different levels of accuracy and computational cost. Within each family, we train a pool of models, compute the ensemble of different combinations of models, and compare these ensembles with the single models in the family.

We denote an ensemble of $n$ image classification models by $\{M_1, \ldots, M_n\}$, where $M_i$ is the $i^{th}$ model. Given an image $x$, $\alpha_i = M_i(x)$ is a vector representing the logits for each class. To ensemble the $n$ models, we compute the mean of logits[1] $\alpha^{\mathrm{ens}} = \frac{1}{n} \sum_i \alpha_i$ and predicts the class for image $x$ by applying argmax to $\alpha^{\mathrm{ens}}$. The total computation of the ensemble is $\mathrm{FLOPs}^{\mathrm{ens}} = \sum_i \mathrm{FLOPs}(M_i)$, where $\mathrm{FLOPs}(\cdot)$ gives the FLOPs of a model.

We show the top-1 accuracy on ImageNet and FLOPs of single models and ensembles in Figure 2. Since there are many possible combinations of models to ensemble, we only show those Pareto optimal ensembles in the figure. We see that ensembles are more cost-effective than large single models, *e.g.*, EfficientNet-B5/B6/B7 and ResNet-152/200. But in the small computation regime, single models outperform ensembles. For example, the ensemble of 2 B5 matches B7 accuracy while using about 50% less FLOPs. However, ensembles use more FLOPs than MobileNetV2 when they have a similar accuracy.

---

[1]We note that the mean of probabilities is a more general choice since logits can be arbitrarily scaled. In our experiments, we observe that they yield similar performance with the mean of logits being marginally better. The findings in our work hold true no matter which choice is used.

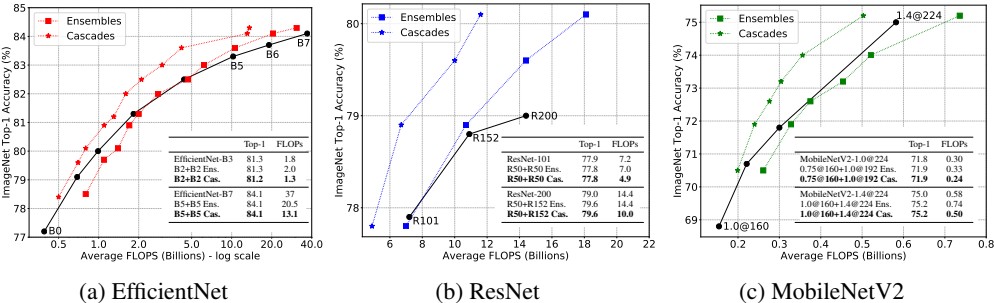

(a) EfficientNet  (b) ResNet  (c) MobileNetV2

Figure 2: Ensembles work well in the *large* computation regime and cascades show benefits in *all* computation regimes. These cascades are directly converted from ensembles without optimizing the choice of models (see Sec. 4). Black dots represent single models. **Ensembles:** Ensembles are more cost-effective than large single models, *e.g.*, EfficientNet-B5/B6/B7 and ResNet-152/200. **Cascades:** Converting ensembles to cascades significantly reduces the FLOPs without hurting the full ensemble accuracy (each star is on the left of a square).

A possible explanation of why model ensembles are more powerful at large computation than at small computation comes from the perspective of bias-variance tradeoff. Large models usually have small bias but large variance, where the variance term dominates the test error. Therefore, ensembles are beneficial at large computation as they can reduce the variance in prediction (Breiman, 1996). For small models, the bias term dominates the test error. Ensembles can reduce the variance, but this cannot compensate the fact that the bias of small models is large. Therefore, ensembles are less powerful at small computation.

Our analysis indicates that instead of using a large model, one should use an ensemble of multiple relatively smaller models, which would give similar performance but with fewer FLOPs. In practice, model ensembles can be easily parallelized (*e.g.*, using multiple accelerators), which may provide further speedup for inference. Moreover, often the total training cost of an ensemble is much lower than that of an equally accurate single model (see appendix for more details).

## 4 FROM ENSEMBLES TO CASCADES

In the above we have identified the scenarios where ensembles outperform or underperform single models. Specifically, ensembles are not an ideal choice when only a small amount of computation is allowed. In this section, we show that by simply converting an ensemble to a cascade, one can significantly reduce the computation and outperform single models in all computation regimes.

Applying an ensemble is wasteful for easy examples where a subset of models will give the correct answer. Cascades save computation via early exit - potentially stopping and outputting an answer before all models are used. The total computation can be substantially reduced if we accurately determine when to exit from cascades. For this purpose, we need a function to measure how likely a prediction is correct. This func-

---

**Algorithm 1** Cascades

**Input**: Models $\{M_i\}$. Thresholds $\{t_i\}$. Test image $x$.
**for** $k = 1, 2, \ldots, n$ **do**
$\quad \alpha^{\mathrm{cas}} = \frac{1}{k}\sum_{i=1}^{k} \alpha_i = \frac{1}{k}\sum_{i=1}^{k} M_i(x)$
$\quad \mathrm{FLOPs}^{\mathrm{cas}} = \sum_{i=1}^{k} \mathrm{FLOPs}(M_i)$
$\quad$ Early exit if confidence score $g(\alpha^{\mathrm{cas}}) \geq t_k$
**end for**
Return $\alpha^{\mathrm{cas}}$ and $\mathrm{FLOPs}^{\mathrm{cas}}$

---

tion is termed *confidence* (more details in Sec. 4.1). A formal procedure of cascades is provided in Algorithm 1. Note that our cascades also average the predictions of the models having been used so far. So for examples where all models are used, the cascade effectively becomes an ensemble.

### 4.1 CONFIDENCE FUNCTION

Let $g(\cdot)\colon \mathbb{R}^N \to \mathbb{R}$ be the confidence function, which maps maps predicted logits $\alpha$ to a confidence score. The higher $g(\alpha)$ is, the more likely the prediction $\alpha$ is correct. Previous work (Huang et al., 2017b; Streeter, 2018) tried several simple metrics to indicate the prediction confidence, such

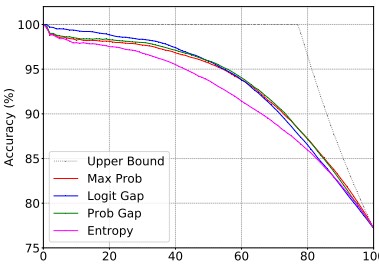 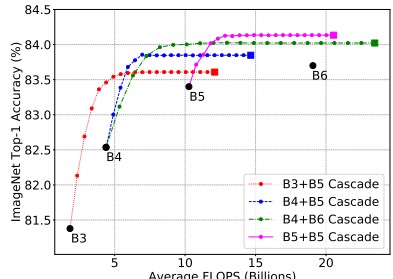

Figure 3: Different metrics for the confidence function. For a EfficientNet-B0 model, we select the top-$k\%$ validation images with highest confidence scores and compute the classification accuracy within the selected images. The higher the accuracy is at a certain $k$, the better the confidence metric is. All the metrics perform similarly in estimating how likely a prediction is correct.

Figure 4: Cascades with different confidence thresholds. Each black dot is a single model and each square is an ensemble of models. Each colored dot represents a cascade with a specific $t_1 (0 \leq t_1 \leq 1)$. As $t_1$ increases from 0 to 1, the cascade uses more and more computation and changes from a single model (first model in the cascade; $t_1 = 0$) to the ensemble ($t_1 = 1$).

as the the maximum probability in the predicted distribution, the gap between the top-2 logits or probabilities, and the (negative) entropy of the distribution. As shown in Figure 3, all the metrics demonstrate reasonably good performance on measuring the confidence of a prediction, *i.e.*, estimating how likely a prediction is correct (see appendix for more details and results). In the following experiments, we adopt the maximum probability metric, *i.e.*, $g(\alpha) = \max(\text{softmax}(\alpha))^2$.

For a cascade of $n$ models $\{M_i\}$, we also need $(n-1)$ thresholds $\{t_i\}$ on the confidence score, where we use $t_i$ to decide whether a prediction is confident enough to exit after applying model $M_i$ (see Algorithm 1). As we define $g(\cdot)$ as the maximum probability, $t_i$ is in $[0, 1]$. A smaller $t_i$ indicates more images will be passed to the next model $M_{i+1}$. A cascade will reduce to an ensemble if all the thresholds $\{t_i\}$ are set to 1. $t_n$ is unneeded, since the cascade will stop after applying the last model $M_n$, no matter how confident the prediction is.

We can flexibly control the trade-off between the computation and accuracy of a cascade through thresholds $\{t_i\}$. To understand how the thresholds influence a cascade, we visualize several 2-model cascades in Figure 4. For each cascade, we sweep $t_1$ from 0 and 1 and plot the results. Note that all the curves in Figure 4 have a plateau, indicating that we can significantly reduce the average FLOPs without hurting the accuracy if $t_1$ is properly chosen. We select the thresholds $\{t_i\}$ on held-out validation images according to the target FLOPs or validation accuracy. In practice, we find such thresholds via grid search. Note that the thresholds are determined after all models are trained. We only need the logits of validation images to determine $\{t_i\}$, so computing the cascade performance for a specific choice of thresholds is fast, which makes grid search computationally possible.

## 4.2 CONVERTING ENSEMBLES TO CASCADES

For each ensemble in Figure 2, we convert it to a cascade that uses the same set of models. During conversion, we set the confidence thresholds such that the cascade performs similar to the ensemble while the FLOPs are minimized. By design in cascades some inputs incur more FLOPs than others. So we report the average FLOPs computed over all images in the test set.

We see that cascades consistently use less computation than the original ensembles and outperform single models in all computation regimes and for all architecture families. Taking 2 EfficientNet-B2 as an example (see Figure 2a), the ensemble initially obtains a similar accuracy to B3 but uses more FLOPs. After converting this ensemble to a cascade, we successfully reduce the average FLOPs to 1.3B (1.4x speedup over B3) and still achieve B3 accuracy. Cascades also outperform small MobileNetV2 models in Figure 2c.

---

[2]As a side observation, when analyzing the confidence function, we notice that models in our experiments are often slightly underconfident. This contradicts the common belief that deep neural networks tend to be overconfident (Guo et al., 2017). Please see appendix for more details.

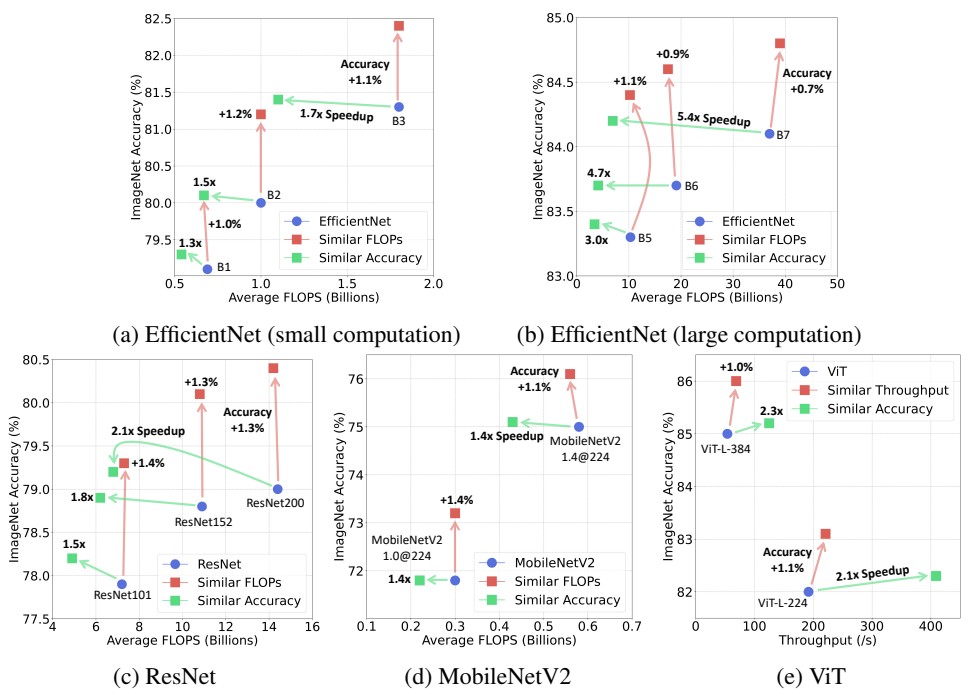

Figure 5: Cascades of EfficientNet, ResNet, MobileNetV2 or ViT models on ImageNet. Compared with single models, cascades can obtain a higher accuracy with similar cost (red squares) or achieve a significant speedup while being equally accurate (green squares; *e.g.*, 5.4x speedup for B7). **The benefit of cascades generalizes to all four architecture families and all computation regimes.** Numerical results are also available in Table 13&14 in appendix.

## 5    MODEL SELECTION FOR BUILDING CASCADES

The cascades in Figure 2 do not optimize the choice of models and directly use the set of models in the original ensembles. For best performance, we show that one can design cascades to match a specific target FLOPs or accuracy by selecting models to be used in the cascade.

Let $\mathcal{M}$ be the set of available models, *e.g.*, models in the EfficientNet family. Given a target FLOPs $\beta$, we select $n$ models $M = \{M_i \in \mathcal{M}\}$ and confidence thresholds $T = \{t_i\}$ by solving the following problem:

$$\max_{\{M_i \in \mathcal{M}\}, \{t_i\}} \text{Accuracy}\left(\mathcal{C}\left(M, T\right)\right)$$
$$s.t. \quad \text{FLOPs}\left(\mathcal{C}\left(M, T\right)\right) \leq \beta, \tag{1}$$

where $\mathcal{C}(M, T)$ is the cascade of models $\{M_i\}$ with thresholds $\{t_i\}$, Accuracy($\cdot$) gives the validation accuracy of a cascade, and FLOPs($\cdot$) gives the average FLOPs. Similarly, we can also build a cascade to match a target validation accuracy $\gamma$ by minimizing FLOPs $\left(\mathcal{C}\left(M, T\right)\right)$ with the constraint Accuracy $\left(\mathcal{C}\left(M, T\right)\right) \geq \gamma$.

Note that this optimization is done after all models in $\mathcal{M}$ were independently trained. The difficulty of this optimization depends on the size of $\mathcal{M}$ and the number of models in the cascade $n$. The problem will be challenging if $|\mathcal{M}|$ or $n$ is large. In our case, $|\mathcal{M}|$ and $n$ are not prohibitive, *e.g.*, $|\mathcal{M}| = 8$ and $n \leq 4$ for EfficientNet family. We are therefore able to solve the optimization problem with exhaustive search. See appendix for more details.

### 5.1    TARGETING FOR A SPECIFIC FLOPs OR ACCURACY

For each single EfficientNet, ResNet or MobileNetV2, we search for a cascade to match its FLOPs (red squares in Figure 5a-5d) or its accuracy (green squares in Figure 5a-5d). Notably, in addition to convolutional networks, we also consider a Transformer architecture – ViT (Dosovitskiy et al., 2021). We build a cascade of ViT-Base and ViT-Large to match the cost or accuracy of ViT-Large (Figure 5e). For ViT, we measure the speedup in throughput (more details on throughput below).

Table 1: Online latency measured on TPUv3. Compared with Efficient-B6 or B7, our cascade achieves a 3.8x or 5.5x reduction in latency respectively.

|  | Top-1 (%) | Latency (ms) | Speedup |
|---|---|---|---|
| B6 | 83.7 | 57.1 | |
| Cascade[*] | 83.7 | **15.1** | **3.8x** |
| B7 | 84.1 | 126.6 | |
| Cascade[*] | 84.2 | **23.2** | **5.5x** |

[*] The cascade that matches B6 or B7 accuracy in Fig. 5b.

Table 2: Offline throughput (images processed per second) measured on TPUv3. Compared with Efficient-B6 or B7, our cascade achieves a 3.0x or 3.5x increase in throughput respectively.

|  | Top-1 (%) | Throughput (/s) | Speedup |
|---|---|---|---|
| B6 | 83.7 | 138 | |
| Cascade[*] | 83.7 | **415** | **3.0x** |
| B7 | 84.1 | 81 | |
| Cascade[*] | 84.2 | **280** | **3.5x** |

[*] The cascade that matches B6 or B7 accuracy in Fig. 5b.

When building cascades, we consider all networks in the same family as the set of available models. The same model type is allowed to be used for multiple times in a cascade but they will be different models trained separately. For ImageNet experiments, the search is conducted on a small set of held-out training images and cascades are evaluated on the original validation set. We provide more experimental details in appendix.

Results in Figure 5 further substantiate our finding that cascades are more efficient than single models in all computation regimes. For small models, we can outperform MobileNetV2-1.0@224 by 1.4% using equivalent FLOPs. For large models, we can obtain 2.3x speedup over ViT-L-384 and 5.4x over EfficientNet-B7 while matching their accuracy.

To better understand how a cascade works, we compute the percentage of images that exit from the cascade at each stage. The cascade above that matches B7 accuracy contains four models: [B3, B5, B5, B5]. In this cascade, 67.3% images only consume the cost of B3 and only 5.5% images use all four models. This saves a large amount of computation compared with using B7 for all the images.

**On-device Latency and Throughput.** In the above, we mostly use average FLOPs to measure the computational cost. We now report the latency and throughput of cascades on TPUv3 in Table 1& 2 to confirm that the reduction in FLOPs can translate to the real speedup on hardware.

Cascades are useful for online processing with a fixed batch size 1. Using batch size 1 is sub-optimal for hardware, but it still happens in real-world applications, *e.g.*, mobile phone cameras processing a single image (Wadhwa et al., 2018) or servers that need to rapidly return the result without waiting for enough queries to form a batch. Table 1 shows the average latency of cascades on TPUv3 with batch size 1. Cascades are up to 5.5x faster than single models with comparable accuracy.

Cascades are also useful for offline data processing, where work can be batched to fully utilize the hardware. We can apply the first model in the cascade to all examples, and then select only a subset of examples to apply the second model and so forth. Table 2 reports the throughput (images processed per second) of cascades on TPUv3 via batch processing. Cascades have significantly higher throughput than comparable models. We provide more results in appendix.

**Comparison with NAS.** We also compare with state-of-the-art NAS methods, *e.g.*, Big-NAS (Yu et al., 2020), OFA (Cai et al., 2020) and Cream (Peng et al., 2020), which can find architectures better than EfficientNet. But as shown in Table 3, a simple cascade of EfficientNet without tuning the architecture already outperforms these sophisticated NAS methods. The strong performance and simplicity of cascades should motivate future research to include them as a strong baseline when proposing novel architectures.

Table 3: Comparison with SOTA NAS methods. Cascades outperform novel architectures found by costly NAS methods.

|  | Top-1 (%) | FLOPs (B) |
|---|---|---|
| BigNASModel-L (Yu et al., 2020) | 79.5 | 0.59 |
| OFA$_{Large}$ (Cai et al., 2020) | 80.0 | 0.60 |
| Cream-L (Peng et al., 2020) | 80.0 | 0.60 |
| Cascade[*] | **80.1** | 0.67 |
| BigNASModel-XL (Yu et al., 2020) | 80.9 | 1.0 |
| Cascade[*] | **81.2** | 1.0 |

[*] The cascade that matches B1 or B2 FLOPs in Figure 5a.

## 5.2 GUARANTEE ON WORST-CASE FLOPS

Up until now we have been measuring the computation of a cascade using the average FLOPs across all images. But for some images, it is possible that all the models in the cascade need to be applied. In this case, the average FLOPs cannot fully indicate the computational cost of a cascade. For example, the cascade that matches B5 or B6 accuracy in Figure 5b has higher worst-case FLOPs

Table 4: Cascades can be built with a guarantee on worst-case FLOPs. We use 'with' or 'w/o' to indicate whether a cascade can provide such a guarantee or not. Cascades with such a guarantee are assured to use fewer FLOPs than single models in the worst-case scenario, and also achieve a considerable speedup in average-case FLOPs.

| | Top-1 (%) | Average-case FLOPs (B) | Worst-case FLOPs (B) | Average-case Speedup | | Top-1 (%) | Average-case FLOPs (B) | Worst-case FLOPs (B) | Average-case Speedup |
|---|---|---|---|---|---|---|---|---|---|
| B5 | 83.3 | 10.3 | 10.3 | | B6 | 83.7 | 19.1 | 19.1 | |
| w/o[*] | 83.4 | 3.4 | 14.2 | 3.0x | w/o[*] | 83.7 | 4.1 | 25.9 | 4.7x |
| with | 83.3 | **3.6** | **9.8** | **2.9x** | with | 83.7 | **4.2** | **15.0** | **4.5x** |

[*] Cascades from Figure 5b.

Table 5: Self-cascades. In the column of self-cascades, the two numbers represent the two resolutions $r_1$ and $r_2$ used in the cascade. Self-cascades use fewer FLOPs than comparable single models.

| EfficientNet | Top-1 (%) | FLOPs (B) | Self-cascades | Top-1 (%) | FLOPs (B) | Speedup |
|---|---|---|---|---|---|---|
| B2 | 80.0 | 1.0 | B1-240-300 | 80.1 | **0.85** | **1.2x** |
| B3 | 81.3 | 1.8 | B2-260-380 | 81.3 | **1.6** | **1.2x** |
| B4 | 82.5 | 4.4 | B3-300-456 | 82.5 | **2.7** | **1.7x** |
| B5 | 83.3 | 10.3 | B4-380-600 | 83.4 | **6.0** | **1.7x** |
| B6 | 83.7 | 19.1 | B5-456-600 | 83.8 | **12.0** | **1.6x** |
| B7 | 84.1 | 37 | B6-528-600 | 84.1 | **22.8** | **1.6x** |

than the comparable single models (see 'w/o' in Table 4). Therefore, we now consider worst-case FLOPs of a cascade, the sum of FLOPs of all models in the cascade.

We can easily find cascades with a guarantee on worst-case FLOPs by adding one more constraint: $\sum_i \text{FLOPs}(M_i) \leq \beta^{\text{worst}}$, where $\beta^{\text{worst}}$ is the upper bound on the worst-case FLOPs of the cascade. With the new condition, we re-select models in the cascades to match of accuracy of B5 or B6. As shown in Table 4, compared with single models, the new cascades achieve a significant speedup in average-case FLOPs and also ensure its worst-case FLOPs are smaller. The new cascades with the guarantee on worst-case FLOPs are useful for applications with strict requirement on response time.

## 6 SELF-CASCADES

Cascades typically contain multiple models. This requires training multiple models and combining them after training. What about when only one model is available? We demonstrate that one can convert a single model into a cascade by passing the same input image at different resolutions to the model. Here, we leverage the fact that resizing an image to a higher resolution than the model is trained on often yields a higher accuracy (Touvron et al., 2019) at the cost of more computation. We call such cascades as "self-cascades" since these cascade only contain the model itself.

Given a model $M$, we build a 2-model cascade, where the first model is applying $M$ at resolution $r_1$ and the second model is applying $M$ at a higher resolution $r_2(r_2 > r_1)$. We build self-cascades using EfficientNet models. Since each EfficientNet is defined with a specific resolution (*e.g.*, 240 for B1), we set $r_1$ to its original resolution and set $r_2$ to a higher resolution. We set the confidence threshold such that the self-cascade matches the accuracy of a single model.

Table 5 shows that self-cascades easily outperform single models, *i.e.*, obtaining a similar accuracy with fewer FLOPs. Table 5 also suggests that if we want to obtain B7 accuracy, we can train a B6 model and then build a self-cascade, which not only uses much fewer FLOPs during inference, but also takes much shorter time to train.

Self-cascades provide a way to convert one single model to a cascade which will be more efficient than the original single model. The conversion is almost free and does not require training any additional models. They are useful when one does not have resources to train additional models or the training data is unavailable (*e.g.*, the model is downloaded).

## 7 APPLICABILITY BEYOND IMAGE CLASSIFICATION

We now demonstrate that the benefit of cascades generalizes beyond image classification.

### 7.1 VIDEO CLASSIFICATION

Similar to image classification, a video classification model outputs a vector of logits over possible classes. We use the same procedure as above to build cascades of video classification models.

Table 6: Cascades of X3D models on Kinetics-600. We outperform X3D-XL by 1.2%.

| | Single Models | | Cascades - Similar FLOPs | | | Cascades - Similar Accuracy | | |
|---|---|---|---|---|---|---|---|---|
| | Top-1 (%) | FLOPs (B) | Top-1 (%) | FLOPs (B) | ΔTop-1 | Top-1 (%) | FLOPs (B) | Speedup |
| X3D-M | 78.8 | 6.2 × 30 | **80.3** | 5.7 × 30 | **1.5** | 79.1 | **3.8 × 30** | **1.6x** |
| X3D-L | 80.6 | 24.8 × 30 | **82.7** | 24.6 × 30 | **2.1** | 80.8 | **7.9 × 30** | **3.2x** |
| X3D-XL | 81.9 | 48.4 × 30 | **83.1** | 38.1 × 30 | **1.2** | 81.9 | **13.0 × 30** | **3.7x** |

We consider the X3D (Feichtenhofer, 2020) architecture family for video classification, which is the state-of-the-art in terms of both the accuracy and efficiency. The X3D family contains a series of models of different sizes. Specifically, we build cascades of X3D models to match the FLOPs or accuracy of X3D-M, X3D-L or X3D-XL on Kinetics-600 (Carreira et al., 2018).

The results are summarized in Table 6, where cascades significantly outperform the original X3D models. Following X3D Feichtenhofer (2020), '×30' in Table 6 means we sample 30 clips from each input video during evaluation (see appendix for more details). Our cascade outperforms X3D-XL, a state-of-the-art video classification model, by 1.2% while using fewer average FLOPs. Our cascade can also match the accuracy of X3D-XL with 3.7x fewer average FLOPs.

## 7.2 SEMANTIC SEGMENTATION

In semantic segmentation, models predict a vector of logits for each pixel in the image. This differs from image classification, where the model makes a single prediction for the entire image. We therefore revisit the confidence function definition to handle such dense prediction tasks.

Table 7: Cascades of DeepLabv3 models on Cityscapes.

| | mIoU | FLOPs (B) | Speedup |
|---|---|---|---|
| ResNet-50 | 77.1 | 348 | - |
| ResNet-101 | 78.1 | 507 | - |
| Cascade - full | 78.4 | 568 | 0.9x |
| Cascade - $s = 512$ | 78.1 | 439 | 1.2x |
| Cascade - $s = 128$ | 78.2 | **398** | **1.3x** |

Similar to before, we use the maximum probability to measure the confidence of the prediction for a single pixel $p$, *i.e.*, $g(\alpha_p) = \max(\text{softmax}(\alpha_p))$, where $\alpha_p$ is the predicted logits for pixel $p$. Next, we need a function $g^{\text{dense}}(\cdot)$ to rate the confidence of the dense prediction for an image, so that we can decide whether to apply the next model to this image based on this confidence score. For this purpose, we define $g^{\text{dense}}(\cdot)$ as the average confidence score of all the pixels in the image: $g^{\text{dense}}(R) = \frac{1}{|R|} \sum_{p \in R} g(\alpha_p)$, where $R$ represents the input image.

In a cascade of segmentation models, we decide whether to pass an image $R$ to the next model based on $g^{\text{dense}}(\cdot)$. Since the difficulty to label different parts in one image varies significantly, *e.g.*, roads are easier to segment than traffic lights, making a single decision for the entire image can be inaccurate and leads to a waste of computation. Therefore, in practice, we divide an image into grids and decide whether to pass each grid to the next model separately.

We conduct experiments on Cityscapes (Cordts et al., 2016) and use mean IoU (mIoU) as the metric. We build a cascade of DeepLabv3-ResNet-50 and DeepLabv3-ResNet-101 (Chen et al., 2017) and report the reults in Table 7. $s$ is the size of the grid. The full image resolution is 1024×2048, so $s = 512$ means the image is divided into 8 grids. If we operate on the full image level ('full'), the cascade will use more FLOPs than ResNet-101. But if operating on the grid level, the cascade can successfully reduce the computation without hurting the performance. For example, the smaller grid size ('$s = 128$') yields 1.3x reduction in FLOPs while matching the mIoU of ResNet-101.

## 8 CONCLUSION

We show that committee-based models, *i.e.*, model ensembles or cascades, provide a simple complementary paradigm to obtain efficient models without tuning the architecture. Notably, cascades can match or exceed the accuracy of state-of-the-art models on a variety of tasks while being drastically more efficient. Moreover, the speedup of model cascades is evident in both FLOPs and on-device latency and throughput. The fact that these simple committee-based models outperform sophisticated NAS methods, as well as manually designed architectures, should motivate future research to include them as strong baselines whenever presenting a new architecture. For practitioners, committee-based models outline a simple procedure to improve accuracy while maintaining efficiency that only needs off-the-shelf models.

AUTHOR CONTRIBUTIONS

Xiaofang wrote most of the code and paper, and ran most of the experiments. Elad and Yair advised on formulating the research question and plan. Dan generated the predictions of X3D on Kinetics-600. Eric conducted the experiments about the calibration of models. Kris helped in writing the paper and provided general guidance.

ACKNOWLEDGMENTS

The authors would like to thank Alex Alemi, Sergey Ioffe, Shankar Krishnan, Max Moroz, and Matthew Streeter for their valuable help and feedback during the development of this work. Elad and Yair would like to thank Solomonico 3rd for inspiration.

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

## A    ON-DEVICE LATENCY AND THROUGHPUT

We report the on-device latency and throughput of cascades to confirm that the reduction in FLOPs can translate to the real speedup on hardware. The latency or throughput of a model is highly dependent on the batch size. So we consider two scenarios: (1) online processing, where we use a fixed batch size 1, and (2) offline processing, where we can batch the examples.

**Online Processing.** Cascades are useful for online processing with a fixed batch size 1. Using batch size 1 is sub-optimal for the utilization of accelerators like GPU or TPU, but it still happens in some real-world applications, *e.g.*, mobile phone cameras processing a single image (Wadhwa et al., 2018) or servers that need to rapidly return the result without waiting for enough queries to form a batch. We report the average latency of cascades on TPUv3 with batch size 1 in Table 8. Cascades achieve a similar accuracy to single models but use a much smaller average latency to return the prediction.

**Offline Processing.** Cascades are also useful for offline processing of large-scale data. For example, when processing all frames in a large video dataset, we can first apply the first model in the cascade to all frames, and then select a subset of frames based on the prediction confidence to apply following models in the cascade. In this way all the processing can be batched to fully utilize the accelerators. We report the throughput of cascades on TPUv3 in Table 9, which is measured as the number of images processed per second. We use batch size 16 when running models on TPUv3 for the case of offline processing. As shown in Table 9, cascades achieve a much higher throughput than single models while being equally accurate. For clarification, only the throughput of ViT in Table 14 is measured on RTX 3090 while the throughput for other models is measured on TPUv3.

## B    DETAILS OF IMAGENET MODELS

When analyzing the efficiency of ensembles or cascades on ImageNet, we consider four architecture families: EfficientNet (Tan & Le, 2019), ResNet (He et al., 2016), MobileNetV2 (Sandler et al., 2018), and ViT (Dosovitskiy et al., 2021). All the single models are independently trained with their original training procedure. We do not change the training schedule or any other hyper-parameters.

- The EfficientNet family contains 8 architectures (EfficientNet-B0 to B7). We train each architecture separately for 4 times with the official open-source implementation[3] provided by the authors. So, in total there are 32 EfficientNet models.

- For ResNet, we consider 4 architectures (ResNet-50/101/152/200) and train each architecture for 2 times using an open-source TPU implementation[4]. There are 8 ResNet models in total.

- For MobileNetV2, we directly download the pre-trained checkpoints from its official open-source implementation[5]. We use 5 MobileNetV2 models: MobileNetV2-0.75@160, 1.0@160, 1.0@192, 1.0@224, and 1.4@224. Each model is represented in the form of $w@r$, where $w$ is the width multiplier and $r$ is the image resolution.

- For ViT, we directly use the pre-trained checkpoints provided by the Hugging Face Team[6]. We use 4 ViT models: ViT-B-224, ViT-L-224, ViT-B-384, and ViT-L-384.

Training each EfficientNet architecture for 4 times (in total 32 models) may sound computationally expensive. We note that it is unnecessary to train each architecture for 4 times to find a well-performing ensemble or cascade. We train a large pool of EfficientNet models mainly for the purpose of analysis so that we can try a diverse range of model combinations, *e.g.*, the cascade of 4 EfficientNet-B5. We analyze the influence of the size and diversity of the model pool in Sec. E.4.

---

[3]https://github.com/tensorflow/tpu/tree/master/models/official/efficientnet

[4]https://github.com/tensorflow/tpu/tree/master/models/official/resnet

[5]https://github.com/tensorflow/models/tree/master/research/slim/nets/mobilenet

[6]For example, ViT-B-224: https://huggingface.co/google/vit-base-patch16-224

Table 8: Average latency on TPUv3 for the case of online processing with batch size 1. Cascades are much faster than single models in terms of the average latency while being similarly accurate.

| | Top-1 (%) | Latency (ms) | Speedup |
|---|---|---|---|
| B1 | 79.1 | 3.7 | |
| Cascade* | 79.3 | **3.0** | **1.2x** |
| B2 | 80.0 | 5.2 | |
| Cascade* | 80.1 | **3.7** | **1.4x** |
| B3 | 81.3 | 9.7 | |
| Cascade* | 81.4 | **5.9** | **1.7x** |
| B4 | 82.5 | 16.6 | |
| Cascade* | 82.6 | **9.6** | **1.7x** |
| B5 | 83.3 | 27.2 | |
| Cascade* | 83.4 | **14.3** | **1.9x** |
| B6 | 83.7 | 57.1 | |
| Cascade* | 83.7 | **15.1** | **3.8x** |
| B7 | 84.1 | 126.6 | |
| Cascade* | 84.2 | **23.2** | **5.5x** |

\* The cascade that matches the accuracy of EfficientNet-B1 to B7 in Figure 5 or the right column of Table 13.

Table 9: Throughput on TPUv3 for the case of offline processing. Throughput is measured as the number of images processed per second. Cascades achieve a much larger throughput than single models while being equally accurate.

| | Top-1 (%) | Throughput (/s) | Speedup |
|---|---|---|---|
| B1 | 79.1 | 1436 | |
| Cascade* | 79.3 | **1798** | **1.3x** |
| B2 | 80.0 | 1156 | |
| Cascade* | 80.1 | **1509** | **1.3x** |
| B3 | 81.3 | 767 | |
| Cascade* | 81.4 | **1111** | **1.4x** |
| B4 | 82.5 | 408 | |
| Cascade* | 82.6 | **656** | **1.6x** |
| B5 | 83.3 | 220 | |
| Cascade* | 83.4 | **453** | **2.1x** |
| B6 | 83.7 | 138 | |
| Cascade* | 83.7 | **415** | **3.0x** |
| B7 | 84.1 | 81 | |
| Cascade* | 84.2 | **280** | **3.5x** |

\* The cascade that matches the accuracy of EfficientNet-B1 to B7 in Figure 5 or the right column of Table 13.

## C   ENSEMBLES ARE ACCURATE, EFFICIENT, AND FAST TO TRAIN

### C.1   EXPERIMENTAL DETAILS

We analyze the efficiency of ensembles of EfficientNet, ResNet, or MobileNetV2 on ImageNet. For EfficientNet, we consider ensembles of two to four models of either the same or different architectures. Note that we only try different combinations of architectures used in the ensemble,

Table 10: Training time (TPUv3 days) of EfficientNet.

| B0 | B1 | B2 | B3 | B4 | B5 | B6 | B7 |
|----|----|----|----|----|----|-----|-----|
| 9 | 12 | 15 | 24 | 32 | 48 | 128 | 160 |

Table 11: Training time (TPU v3 days) of ensembles. We use the '+' notation to indicate the models in enmsebles. Ensembles are faster than single models in both training and inference while achieve a similar accuracy.

|          | Top-1 (%) | FLOPs (B) | Training |
|----------|-----------|-----------|----------|
| B6       | 83.7      | 19.1      | 128      |
| B3+B4+B4 | 83.6      | 10.6      | 88       |
| B7       | 84.1      | 37        | 160      |
| B5+B5    | 84.1      | 20.5      | 96       |
| B5+B5+B5 | 84.3      | 30.8      | 144      |

but not the combinations of models. For example, when an ensemble contains an EfficientNet-B5, while we have multiple B5 models available, we just randomly pick one but do not try all possible choices. The FLOPs range of ResNet or MobileNetV2 models is relatively narrow compared with EfficientNet, so we only consider ensembles of two models for ResNet and MobileNetV2.

### C.2   TRAINING TIME OF ENSEMBLES

In Sec. 3, we show that ensembles match the accuracy of large single models with fewer inference FLOPs. We now show that the total training cost of an ensemble if often lower than an equally accurate single model.

We show the training time of single EfficinetNet models in Table 10. We use 32 TPUv3 cores to train B0 to B5, and 128 TPUv3 cores to train B6 or B7. All the models are trained with the public official implementation of EfficientNet. We choose the ensemble that matches the accuracy of B6 or B7 and compute the total training time of the ensemble based on Table 10. As shown in Table 11, the ensemble of 2 B5 can match the accuracy of B7 while being faster in both training and inference.

## D   FROM ENSEMBLES TO CASCADES

### D.1   CONFIDENCE FUNCTION

The higher the confidence score $g(\alpha)$ is, the more likely the prediction given by $\alpha$ is correct. In Sec 4.1, we compare different choices for the confidence function in Figure 3. For a specific confidence function, we select the top-$k\%$ images with highest confidence scores. Then we compute the classification accuracy within the selected images. If a higher confidence score indicates that the prediction is more likely to be correct, the accuracy should drop as as $k$ increases.

Figure 3 is generated with a EfficientNet-B0 model trained on ImageNet, where we sweep $k$ from 0 to 100 and compute the accuracy within the selected top-$k\%$ images from the ImageNet validation set. When $k = 100$, all the images are selected so the accuracy is exactly the accuracy of EfficientNet-B0 (77.1%). The 'Upper Bound' curve represents the best possible performance for the metric. It has 100% accuracy when $k \leq 77.1$, *i.e.*, all the selected images are correctly classified. The accuracy starts to drop when $k$ becomes larger, since some misclassified images are inevitably chosen. We observe all the metrics demonstrate reasonably good performance in estimating how likely a prediction is correct, where the entropy performs slightly worse than other metrics.

We also compare the performance of the cascade of ViT-B-224 and ViT-L-224 on ImageNet with different confidence functions in Table 12. For each confidence function, we set the threshold such that the cascade has a similar throughput when using different confidence functions ($\sim$409 images

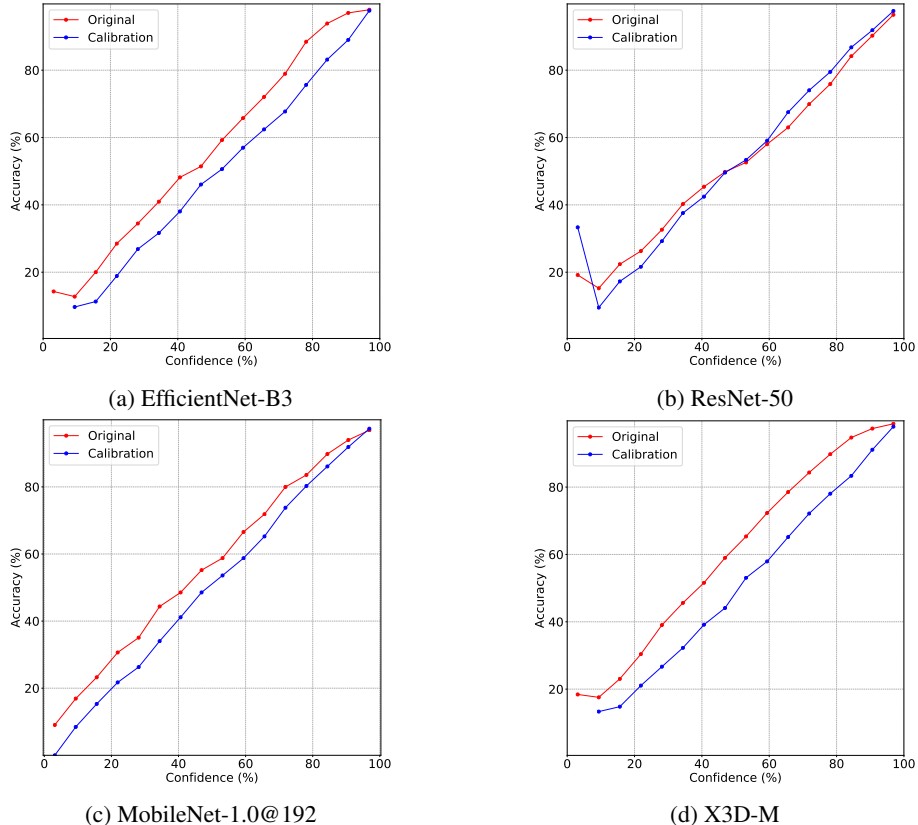

(a) EfficientNet-B3             (b) ResNet-50

(c) MobileNet-1.0@192          (d) X3D-M

Figure 6: Confidence vs. accuracy for EfficientNet-B3, ResNet-50, MobileNet-1.0@192, and X3D-M on their respective dataset. 'Original' refers to the original prediction and 'Calibration' refers to the prediction after calibration. We divide the confidence into several intervals. Then for each interval, we visualize the accuracy of images whose confidence is within this interval. The original prediction is slightly underconfident, *i.e.*, the confidence $p$ is slightly lower than the actual accuracy of images whose prediction confidence is $p$. After calibration, the confidence almost equals to the actual accuracy.

Table 12: Performance of the cascade of ViT-B-224 and ViT-L-224 on ImageNet with different confidence functions. For each confidence function, we set the threshold such that the cascade has a similar throughput when using different confidence functions ($\sim$409 images per second). The table shows that different confidence functions give a similar accuracy.

|             | Top-1 (%) |
| ----------- | --------- |
| Max Prob    | 82.3      |
| Logit Gap   | 82.2      |
| Prob Gap    | 82.3      |
| Entropy Gap | 82.1      |

per second). We observe that the cascade achieves a similar accuracy with different confidence functions.

## D.2 CALIBRATION OF MODELS

As a side observation, when analyzing the confidence function, we notice that models in our experiments are often slightly underconfident, *i.e.*, the confidence $p$ is slightly lower than the actual accuracy of images whose prediction confidence is $p$ (see the 'Original' curve in Figure 6). This observation contradicts the common belief that deep neural networks tend to be overconfident (Guo

et al., 2017). We conjecture this is due to that our models are trained with label smoothing (Szegedy et al., 2016). Here, the confidence of a prediction is defined as the probability associated with the predicted class (Guo et al., 2017), which is equivalent to the maximum probability in the predicted distribution.

A model is considered as calibrated if the confidence of its prediction correctly estimates the true correctness likelihood (neither overconfident nor underconfident). The calibration of models can influence the ensemble performance when we ensemble models via simple averaging. If models in an ensemble are poorly calibrated, overconfident models may dominate the prediction, making those underconfident models useless in the ensemble.

For models in our experiments, we also have tried calibrating them before computing the ensemble performance. To calibrate a model, we use Platt scaling via a learned monotonic calibration network. As shown in the 'Calibration' curve in Figure 6, the calibration improves the connection between the prediction confidence and accuracy, *i.e.*, the confidence after calibration almost equals to the actual accuracy. But we notice that calibrating models only has a small influence on the final ensemble performance in our experiments, which might be because these models are just slightly underconfident before calibration. Therefore, we do not calibrate any model when computing the ensemble in all our experiments.

### D.3    DETERMINE THE THRESHOLDS

Given the $n$ models $\{M_i\}$ in a cascade, we also need $(n-1)$ thresholds $\{t_i\}$ on the confidence score. We can flexibly control the trade-off between the computation and accuracy of a cascade through thresholds $\{t_i\}$.

We determine the thresholds $\{t_i\}$ based on the target FLOPs or accuracy on validation images. In practice, we find such thresholds via grid search, *i.e.*, enumerating all possible combinations for $\{t_i\}$. Note that the thresholds are determined after all models are trained. We only need the logits of validation images to determine $\{t_i\}$, so computing the cascade performance for a specific choice of thresholds is fast, which makes grid search computationally possible. As $t_i$ is a real number, we make sure two trials of $t_i$ are sufficiently different by only considering the percentiles of confidence scores as possible values. When $n > 2$, there might be multiple choices of $\{t_i\}$ that can give the target FLOPs or accuracy. In that case, we choose $\{t_i\}$ that gives the higher accuracy or fewer FLOPs. Many choices for $\{t_i\}$ can be easily ruled out as the FLOPs or accuracy of a cascade changes monotonically with respect to any threshold $t_i$.

In practice, we often want the accuracy of a cascade to match the accuracy of a single model. To do that, we determine the thresholds such that the cascade matches the accuracy of the single model on *validation* images. Such thresholds usually enable the cascade to have a similar *test* accuracy to the single model.

For ImageNet, we randomly select $\sim$25k training images and exclude them from training. We use these held-out training images to determine the confidence thresholds. The final accuracy is computed on the original ImageNet validation set.

## E    MODEL SELECTION FOR BUILDING CASCADES

### E.1    TARGETING FOR A SPECIFIC FLOPS OR ACCURACY

We can build cascades to match a specific FLOPs or accuracy by optimizing the choice of models and confidence thresholds, *e.g.*, solving Eq. 1 when targeting for a specific FLOPs. Note that this optimization is done after all models in $M$ are trained. The optimization complexity is exponential in $|\mathcal{M}|$ and $n$, and the problem will be challenging if $|\mathcal{M}|$ and $n$ are large. In our experiments, $|\mathcal{M}|$ and $n$ are not prohibitive. Therefore, we solve the optimization problem with exhaustive search. One can also use more efficient procedures such as the algorithm described in (Streeter, 2018).

Same as the our analysis of ensembles, we do not search over different models of the same architecture, but only search combination of architectures. Therefore, for EfficientNet, $|\mathcal{M}| = 8$ and $n \leq 4$ and we have in total $4672 = (8^4 + 8^3 + 8^2)$ possible combinations of models. Note that the search is cheap to do as it is conducted after all the models are independently trained. No GPU training is

Table 13: Cascades of EfficientNet, ResNet or MobileNetV2 models on ImageNet. This table contains the numerical results for Figure 5a-5d. **Middle**: Cascades obtain a higher accuracy than single models when using similar FLOPs. **Right**: Cascades achieve a similar accuracy to single models with significantly fewer FLOPs (*e.g.*, 5.4x fewer for B7). **The benefit of cascades generalizes to all three convolutional architecture families and all computation regimes.**

| | Single Models | | Cascades - Similar FLOPs | | | Cascades - Similar Accuracy | | |
|---|---|---|---|---|---|---|---|---|
| | Top-1 (%) | FLOPs (B) | Top-1 (%) | FLOPs (B) | $\Delta$Top-1 | Top-1 (%) | FLOPs (B) | Speedup |
| **EfficientNet** | | | | | | | | |
| B1 | 79.1 | 0.69 | **80.1** | 0.67 | **1.0** | 79.3 | **0.54** | **1.3x** |
| B2 | 80.0 | 1.0 | **81.2** | 1.0 | **1.2** | 80.1 | **0.67** | **1.5x** |
| B3 | 81.3 | 1.8 | **82.4** | 1.8 | **1.1** | 81.4 | **1.1** | **1.7x** |
| B4 | 82.5 | 4.4 | **83.7** | 4.1 | **1.2** | 82.6 | **2.0** | **2.2x** |
| B5 | 83.3 | 10.3 | **84.4** | 10.2 | **1.1** | 83.4 | **3.4** | **3.0x** |
| B6 | 83.7 | 19.1 | **84.6** | 17.5 | **0.9** | 83.7 | **4.1** | **4.7x** |
| B7 | 84.1 | 37 | **84.8** | 39.0 | **0.7** | 84.2 | **6.9** | **5.4x** |
| **ResNet** | | | | | | | | |
| R101 | 77.9 | 7.2 | **79.3** | 7.3 | **1.4** | 78.2 | **4.9** | **1.5x** |
| R152 | 78.8 | 10.9 | **80.1** | 10.8 | **1.3** | 78.9 | **6.2** | **1.8x** |
| R200 | 79.0 | 14.4 | **80.4** | 14.2 | **1.3** | 79.2 | **6.8** | **2.1x** |
| **MobileNetV2** | | | | | | | | |
| 1.0@160 | 68.8 | 0.154 | **69.5** | 0.153 | **0.6** | 69.1 | **0.146** | **1.1x** |
| 1.0@192 | 70.7 | 0.22 | **71.8** | 0.22 | **1.1** | 70.8 | **0.18** | **1.2x** |
| 1.0@224 | 71.8 | 0.30 | **73.2** | 0.30 | **1.4** | 71.8 | **0.22** | **1.4x** |
| 1.4@224 | 75.0 | 0.58 | **76.1** | 0.56 | **1.1** | 75.1 | **0.43** | **1.4x** |

Table 14: Cascades of ViT models on ImageNet. This table contains the numerical results for Figure 5e. 224 or 384 indicates the image resolution the model is trained on. Throughput is measured on NVIDIA RTX 3090. Our cascades can achieve a 1.0% higher accuracy than ViT-L-384 with a similar throughput or achieve a 2.3x speedup over it while matching its accuracy. **The benefit of cascades generalizes to Transformer architectures.**

| | Single Models | | Cascades - Similar Throughput | | | Cascades - Similar Accuracy | | |
|---|---|---|---|---|---|---|---|---|
| | Top-1 (%) | Throughput (/s) | Top-1 (%) | Throughput (/s) | $\Delta$Top-1 | Top-1 (%) | Throughput (/s) | Speedup |
| ViT-L-224 | 82.0 | 192 | **83.1** | 221 | **1.1** | 82.3 | **409** | **2.1x** |
| ViT-L-384 | 85.0 | 54 | **86.0** | 69 | **1.0** | 85.2 | **125** | **2.3x** |

involved in the search. We pre-compute the predictions of each model on a held-out validation set before search. During the search, we try possible models combinations by loading their predictions. We can usually find optimal model combinations within a few CPU hours.

In practice, we first train each EfficientNet model separately for 4 times and pre-compute their predicted logits. Then for each possible combination of models, we load the logits of models and determine the thresholds according to the target FLOPs or accuracy. Finally, we choose the best cascade among all possible combinations. Similar as above, we choose models and thresholds on held-out training images for ImageNet experiments. No images from the ImageNet validation set are used when we select models for a cascade.

For ResNet and MobileNetV2, we only tried 2-model cascades due to their relatively narrow FLOPs range. Therefore, the number of possible model combinations is very small ($< 20$). For ViT, we only tried 2 cascades: ViT-B-224 + ViT-L-224 and ViT-B-384 + ViT-L-384.

## E.2 CASCADES CAN BE SCALED UP

One appealing property of single models is that they can be easily scaled up or down based on the available computational resources one has. We show that such property is also applicable to cascades, *i.e.*, we can scale up a base cascade to respect different FLOPs constraints. This avoids the model selection procedure when designing cascades for different FLOPs, which is required for cascades in Table 13.

Table 15: A Family of Cascades C0 to C7. C0 to C7 significantly outperform single EfficientNet models in all computation regimes. C1 and C2 also compare favorably with state-of-the-art NAS methods, such as BigNAS (Yu et al., 2020), OFA (Cai et al., 2020) and Cream (Peng et al., 2020). This shows that the cascades can also be scaled up or down to respect different FLOPs constraints as single models do. This is helpful for avoiding the model selection procedure when designing cascades for different FLOPs.

| Model | Top-1 (%) | FLOPs (B) | $\Delta$Top-1 | Model | Top-1 (%) | FLOPs (B) | $\Delta$Top-1 |
|---|---|---|---|---|---|---|---|
| **C0** | **78.1** | **0.41** | | **C3** | **82.2** | **1.8** | |
| EfficientNet-B0 | 77.1 | 0.39 | 1.0 | EfficientNet-B3 | 81.3 | 1.8 | 0.9 |
| **C1** | **80.3** | **0.71** | | **C4** | **83.7** | **4.2** | |
| EfficientNet-B1 | 79.1 | 0.69 | 1.2 | EfficientNet-B4 | 82.5 | 4.4 | 1.2 |
| BigNASModel-L | 79.5 | 0.59 | 0.8 | **C5** | **84.3** | **10.2** | |
| OFA$_{Large}$ | 80.0 | 0.60 | 0.3 | EfficientNet-B5 | 83.3 | 10.3 | 1.0 |
| Cream-L | 80.0 | 0.60 | 0.3 | **C6** | **84.6** | **18.7** | |
| **C2** | **81.2** | **1.0** | | EfficientNet-B6 | 83.7 | 19.1 | 0.9 |
| EfficientNet-B2 | 80.0 | 1.0 | 1.2 | **C7** | **84.8** | **32.6** | |
| BigNASModel-XL | 80.9 | 1.0 | 0.3 | EfficientNet-B7 | 84.1 | 37 | 0.7 |

Specifically, we build a 3-model cascade to match the FLOPs of EfficientNet-B0. We call this cascade C0 (see below for details of building C0). Then, simply by scaling up the architectures in C0, we obtain a family of cascades C0 to C7 that have increasing FLOPs and accuracy. The models in C0 are from the EfficientNet family. The results of C0 to C7 in Table 15 show that simply scaling up C0 gives us a family of cascades that consistently outperform single models in all computation regimes. This finding enhances the practical usefulness of cascades as one can select cascades from this family based on available resources, without worrying about what models should be used in the cascade.

**Details of building C0.** The networks in EfficientNet family are obtained by scaling up the depth, width and resolution of B0. The scaling factors for depth, width and resolution are defined as $d = \alpha^\phi$, $w = \beta^\phi$ and $r = \gamma^\phi$, respectively, where $\alpha = 1.2$, $\beta = 1.1$ and $\gamma = 1.15$, as suggested in Tan et al. (Tan & Le, 2019). One can control the network size by changing $\phi$. For example, $\phi = 0$ gives B0, $\phi = 1$ gives B2, and $\phi = 7$ gives B7.

We build a 3-model cascade C0 to match the FLOPs of EfficientNet-B0 by solving Eq. 1 on held-out training images from ImageNet. When building C0, we consider 13 networks from EfficientNet family. As we want C0 to use similar FLOPs to B0, we make sure the 13 networks include both networks smaller than B0 and networks larger than B0. Their $\phi$ are set to -4.0, -3.0, -2.0, -1.0, 0.0, 0.25, 0.5, 0.75, 1.0, 1.25, 1.50, 1.75, 2.0, respectively.

The $\phi$ of the three models in C0 are -2.0, 0.0 and 0.75. Then simply scaling up the architectures in C0, *i.e.*, increasing the $\phi$ of each model in C0, gives us a family of cascades C0 to C7 that have increasing FLOPs and accuracy. The thresholds in C0 to C7 are determined such that their FLOPs are similar to B0 to B7.

### E.3 EXIT RATIOS

To better understand how a cascade works, we compute the exit ratio of the cascade, *i.e.*, the percentage of images that exit from the cascade at each stage. Specifically, we choose the cascades in Table 13&4 that match the accuracy of B1 to B7 and report their exit ratios in Table 16. For all the cascades in Table 16, most images only consume the cost of the first model in the cascade and only a few images have to use all the models. This shows that cascades are able to allocate fewer resources to easy images and explains the speedup of cascades over single models.

### E.4 MODEL POOL ANALYSIS

### E.4.1 NUMBER OF MODELS IN CASCADES

We study the influence of the number of models in cascades on the performance. Concretely, we consider the EfficientNet family and follow the same experimental setup as in Sec. E.1. We sweep

Table 16: Exit ratios of cascades. We use the '+' notation to indicate the models in cascades.

| | Top-1 (%) | FLOPs (B) | Exit Ratio (%) at Each Stage | | | |
| --- | --- | --- | --- | --- | --- | --- |
| | | | Model 1 | Model 2 | Model 3 | Model 4 |
| B1 | 79.1 | 0.69 | | | | |
| B0+B1 | 79.3 | 0.54 | 78.7 | 21.3 | | |
| B2 | 80.0 | 1.0 | | | | |
| B0+B1+B3 | 80.1 | 0.67 | 73.2 | 21.4 | 5.4 | |
| B3 | 81.3 | 1.8 | | | | |
| B0+B3+B3 | 81.4 | 1.1 | 68.0 | 26.4 | 5.7 | |
| B4 | 82.5 | 4.4 | | | | |
| B1+B3+B4 | 82.6 | 2.0 | 67.9 | 15.3 | 16.8 | |
| B5 | 83.3 | 10.3 | | | | |
| B2+B4+B4+B4 | 83.4 | 3.4 | 67.6 | 21.2 | 0.0 | 11.2 |
| B2+B4+B4$^*$ | 83.3 | 3.6 | 57.7 | 26.0 | 16.3 | |
| B6 | 83.7 | 19.1 | | | | |
| B2+B4+B5+B5 | 83.7 | 4.1 | 67.6 | 21.2 | 5.9 | 5.3 |
| B3+B4+B4+B4$^*$ | 83.7 | 4.2 | 67.3 | 16.2 | 10.9 | 5.6 |
| B7 | 84.1 | 37 | | | | |
| B3+B5+B5+B5 | 84.2 | 6.9 | 67.3 | 21.6 | 5.6 | 5.5 |

$^*$ Cascades from Table 4 with a guarantee on worst-case FLOPs.

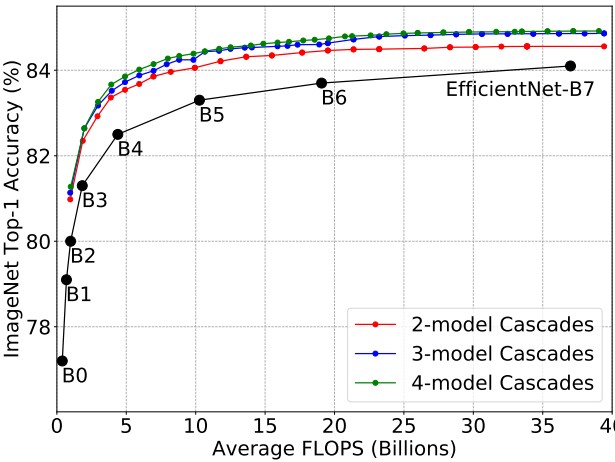

Figure 7: Impact of the number of models in cascades.

the target FLOPs from 1 to 40 and find cascades of 2, 3 or 4 models. As shown in Figure 7, the performance of cascades keeps improving as the number of models increases. We see a big gap between 2-model cascades and 3-model cascades, but increasing the number of models from 3 to 4 demonstrates a diminishing return.

As mentioned above, for EfficientNet cascades, we tried in total $4672 = (8^4 + 8^3 + 8^2)$ possible combinations of models. Since 3-model cascades can obtain very close performance to 4-model cascades, one could try much fewer combinations to obtain similar results.

## E.5    SIZE OF THE MODEL POOL

As mentioned in Sec. B, we train each EfficientNet architecture for 4 times so that we can try a diverse range of model combinations. We now empirically show that naively adding more models

Table 17: Max, min, mean, and standard deviation of the performance of 8 single B5 models, 28 possible 2-B5 ensembles, and 56 possible 2-B5 cascades.

|  | max | min | mean | std |
|---|---|---|---|---|
| **Single Model** | | | | |
| Accuracy (%) | 83.40 | 83.29 | 83.34 | 0.04 |
| **2-B5 Ensembles** | | | | |
| Accuracy (%) | 84.18 | 83.97 | 84.10 | 0.05 |
| **2-B5 Cascades** | | | | |
| Accuracy (%) | 84.17 | 83.96 | 84.09 | 0.05 |
| FLOPs (B) | 13.35 | 12.32 | 12.62 | 0.29 |

Table 18: Cascades of models of same architectures vs. Cascades of models of different architectures. The '+' notation indicates the models used in cascades.

|  | Top-1 (%) | FLOPs (B) | Speedup |
|---|---|---|---|
| B4 | 82.5 | 4.4 | |
| B3+B3+B3 | 82.6 | 2.7 | 1.6x |
| B1+B3+B4 | 82.6 | **2.0** | **2.2x** |
| B5 | 83.3 | 10.3 | |
| B4+B4 | 83.3 | 5.1 | 2.1x |
| B2+B4+B4+B4 | 83.4 | **3.4** | **3.0x** |
| B6 | 83.7 | 19.1 | |
| B4+B4+B4 | 83.8 | 6.0 | 3.2x |
| B2+B4+B5+B5 | 83.7 | **4.1** | **4.7x** |
| B7 | 84.1 | 37 | |
| B5+B5 | 84.1 | 13.1 | 2.8x |
| B3+B5+B5+B5 | 84.2 | **6.9** | **5.4x** |

of the same architecture to the pool only has a small influence on the performance of ensembles or cascades.

We train 8 EfficientNet-B5 models separately and build 2-B5 ensembles or cascades using any two of these models. The FLOPs of these 2-B5 ensembles are the same (20.5B). For each cascade, we tune the confidence threshold such that the cascade achieves a similar accuracy to the full ensemble. We show the max, min, mean, and standard deviation of the performance of these different ensembles or cascades in Table 17 and observe that the performance variation is small. Therefore, we conclude that adding more models of the same architecture only has modest influence on the performance.

### E.5.1 DIVERSITY OF THE MODEL POOL

We study the influence of the diversity of architectures in the model pool on the performance. We compare cascades of models of same architectures and cascades of models of different architectures in Tables 18. As shown in Table 18, while cascades of same-architecture models can already significantly reduce the FLOPs compared with a similarly accurate single model, adding more variations in the architecture can significantly improve the performance of cascades.

## F APPLICABILITY BEYOND IMAGE CLASSIFICATION

### F.1 VIDEO CLASSIFICATION

We conduct video classification on Kinetics-600 (Carreira et al., 2018). Following X3D (Feichtenhofer, 2020), we sample 30 clips from each input video when evaluating X3D models on Kinetics-

600. The 30 clips are the combination of 10 uniformly sampled temporal crops and 3 spatial crops. The final prediction is the mean of all individual predictions.

## F.2 SEMANTIC SEGMENTATION

**Confidence Function.** We notice that many pixels are unlabeled in semantic segmentation datasets, *e.g.*, Cityscapes (Cordts et al., 2016), and are ignored during training and evaluation. These unlabeled pixels may introduce noise when we average the confidence score of all the pixels. To filter out unlabeled pixels in the image, we only consider pixels whose confidence is higher than a preset threshold $t^{\text{unlab}}$. So we update the definition of $g^{\text{dense}}(\cdot)$ as follows: $g^{\text{dense}}(R) = \frac{1}{|R'|} \sum_{p \in R'} g(\alpha_p)$, where $R' = \{p \mid g(\alpha_p) > t^{\text{unlab}}, p \in R\}$.

**Experimental Details.** We conduct experiments on the Cityscapes (Cordts et al., 2016) dataset, where the full image resolution is 1024×2048. We train DeepLabv3 (Chen et al., 2017) models on the train set of Cityscapes and report the mean IoU (mIoU) over classes on the validation set. The threshold $t^{\text{unlab}}$ to filter out unlabeled pixels is set to 0.5. For DeepLabv3-ResNet-50 or DeepLabv3-ResNet-101, we follow the original architecture of ResNet-50 or ResNet-101, except that the first 7x7 convolution is changed to three 3x3 convolutions (see `resnet_v1_beta` in the official DeepLab imeplmentation[7]).

---

[7] `https://github.com/tensorflow/models/blob/master/research/deeplab/core/resnet_v1_beta.py`

