# OpenReview forum: "Wisdom of Committees: An Overlooked Approach To Faster and More Accurate Models"
_ICLR.cc/2022/Conference — ICLR 2022 Poster_

### Official Review · Reviewer_MpvP · 2021-10-29

**Correctness:** 3
**Technical Novelty And Significance:** 2
**Empirical Novelty And Significance:** 3
**Recommendation:** 6
**Confidence:** 3

**Main Review:**

**Pros:**

+ the study of the efficiency-accuracy trade-off in shallow network compositions is a relevant topic;
+ the empirical study on ImageNet is thorough and convincing;
+ nice extensions to NAS-based models and related tasks of video recognition and semantic segmentation;
+ generally very thoughtful experimental setups.

**Cons:**

> Significance: While I appreciate the thoroughness of the study, I have difficulties recognising the significance of the new insight gauged w.r.t. prior work. Compared to (Kondratyuk et al., 2020; Lobacheva et al., 2020), this work simply augments those studies to larger-scale classification, video recognition and semantic segmentation, with ImageNet experiments being by far the dominant content (7 out of 8 pages).

> Reproducibility: This work would be likely very hard to reproduce. The text does not provide any training details (e.g. the training duration, the learning rate and schedule, regularisation, etc.).
Perhaps more importantly, how were the ensemble models trained? Was there any bootstrapping used (e.g. see Ilg et al., (2018)), diversity-encouraging loss terms (e.g., Lee et al., (2016)) or snapshot ensembling? Why or why not?
Related to both the significance and reproducibility: I wonder how much deviation is there on the evaluation metrics (e.g., of the top-1 accuracy) in ensemble/cascade models?

> Choice of the baseline and metrics: 1) I wonder how cascades/ensembles fare against the more efficient types of temporal ensembling (e.g. SWA from Izmailov et al., (2019))? 2) I’m curious what happens to uncertainty estimates. The choice of the max-based confidence function is interesting, but why is this a reasonable choice? The observed improvement in accuracy that it provides may come at a cost of deteriorated calibration; this needs to be verified.

> Clarity. I find it rather unusual to seek ensemble configurations satisfying a particular value of accuracy or the number of FLOPs. It seems only to introduce additional experimental setups that deliver little new insight; there is unnecessary complexity of constraining the ensemble/cascade set, and it’s not always clear which models end up in the composition. I may be overlooking something, but the simple accuracy-efficiency plots (like Fig. 1 or Fig. 2) are already sufficient to draw the same conclusions and are simpler to interpret.

Minor:
- E1 the missing x30 factor in Tab. 8 is important, even though it does not change the ranking.
- Semantic segmentation: Why is it reasonable to change 7x7 convolution to 3x3?
- A number of typos (e.g. “cost-effectiev” on p. 4; “This methods does not reqire” on p. 2).
- “we only show those Pareto optimal ensembles in the figure” needs more explanation. Which models/configurations were evaluated but were not Pareto optimal?


**Post-rebuttal comment**\
I thank the authors for the clarifications. I recognise that the work provides significant empirical evidence (indicated in some previous findings) that compositions of shallower models may be both more accurate and more efficient. I also acknowledge that the revision now includes important details for reproducibility. However, a few things could be improved:

- I agree with the other reviewers that the clarity of presentation should be improved significantly. For example, I find it bizarre to refer to Fig. 2 without introducing the architectures (and justifying their choice), how they were trained and selected for composition (i.e. the cascades are introduced only much later), how the hyperparameters are chosen (e.g. the thresholds), etc. These points are discussed in the paper, but either much later or in different contexts.
- given the available computational resources and that most of the empirical advantages come from ImageNet, it would be interesting to verify these results on another classification dataset (e.g. Open Images, Pascal 2012 for multi-label classification, or at least CIFAR-100).
- model/ensemble calibration needs more analysis. As highlighted in the paper (and in the rebuttal), the calibration of the models used contradicts the view on the networks being overconfident. However, the work makes no effort to explain this finding. Is it an artefact of the dataset? The training schedule? How does it affect the calibration of the cascades/ensembles?
- minor: there are still quite a number of typos.

Nevertheless, I upgrade my score, since the remaining points can be addressed in the camera-ready.

**Summary Of The Paper:**

The work provides an empirical accuracy-efficiency comparison of model ensembles and cascades of shallow models against single deeper models. The main finding, which supports previous results, is rather interesting: compositions of shallow models tend to provide better efficiency-accuracy trade-off than single deep models. This finding has been extended here to the ImageNet classification task with three architecture families (ResNet, EfficientNet and MobileNet), with further examples on video recognition and semantic segmentation.

**Summary Of The Review:**

I find that this work provides a thorough study of the efficiency-accuracy trade-off offered by ensembles of shallow models w.r.t. single deeper models. However, I have difficulties recognising the significance of the new insights this study delivers in light of the previous work (e.g. Lobacheva et al., (2020))

---

> ### Author Response · Authors · 2021-11-22
> **Response to Clarity and Minor Comments**
>
> ### Q: Why do we need to search the ensemble/cascade configurations satisfying a particular accuracy of FLOPs?
> **Answer:**
>
> We search for cascade configurations based on a target accuracy or FLOPs because
> * (1) When the target changes, the optimal cascade config changes. For example, among the several 2-model cascades in Figure 4, the best cascades are different for different FLOPs.
>  * (2) The search is cheap to do and guarantees that an optimal configuration is found.
>      * The search is cheap to do as it is conducted after all the models are independently trained. No GPU training is involved in the search. We pre-compute the predictions of each model on a held-out validation set before search. During the search, we try possible models collections by loading their predictions. We can usually find optimal model collections within a few CPU hours.
>
> ### Q:  I may be overlooking something, but the simple accuracy-efficiency plots (like Fig. 1 or Fig. 2) are already sufficient
> **Answer:**
> * Figure 1 is a summary of our results, where many data points are from Table 1, which is obtained via searching for the optimal cascade configuration.
> * Yes, Fig 2 already demonstrates the superior efficiency of ensembles / cascades. But conducting the search can still give a further significant boost. For example, Figure 2 shows that a B5+B5 cascade can speed up B7 by 2.8x (37B v.s. 13.1B). But as shown in Table 1, the search helps us find a much better cascade (B3+B5+B5+B5) that can speed up B7 by 5.4x  while being similarly accurate.
>
> ### Q: It’s not always clear which models end up in the composition.
> **Answer:**
> * To better understand how a cascade works, in Table 16 we provide the models being used in the cascade and compute the percentage of images that exit from the cascade at each stage.
> * For example, the cascade above that matches B7 accuracy contains four models: [B3, B5, B5, B5]. In this cascade, 67.3% of images only consume the cost of B3 and only 5.5% of images use all four models.
>
>
> ### Q: The missing x30 factor in Tab. 8 is important, even though it does not change the ranking.
> **Answer:**
> * We have updated Table 8 and included the “x30” factor.
>
>
> ### Q: Semantic segmentation: Why is it reasonable to change 7x7 convolution to 3x3?
> **Answer:**
> * We are using the resnet_v1_beta in the official DeepLab implementation (https://github.com/tensorflow/models/blob/master/research/deeplab/core/resnet_v1_beta.py). This is the change made by the authors as they find this empirically gives better performance.
>
>
> ### Q: Which models/configurations were evaluated but were not Pareto optimal?
> **Answer:**
> * For example, B4+B6 ensemble is not Pareto optimal compared to B5+B5 ensemble. B5+B5 ensembles can achieve slightly higher accuracy than B4+B6 (84.1% vs 84.0) but with fewer FLOPs (20.5B vs 23.5). Therefore, we only show B5+B5 in the figure and do not include B4+B6.

---

> ### Author Response · Authors · 2021-11-22
> **Response to Choice of the baseline and metrics**
>
> ### Q:  How cascades/ensembles fare against the more efficient types of temporal ensembling (e.g. SWA from Izmailov et al., (2019))?
> **Answer:**
> * Our contribution is to show that **even the most simplistic method to build ensembles or cascades can achieve superior efficiency**. So, we intentionally chose the simplest possible method and did not explore other sophisticated techniques.
> * Since there might be differences in the training pipeline between ours and SWA, it is unfair to directly compare the absolute accuracy (we actually have higher absolute accuracy). For a fair comparison, we compare the relative improvement over the original single model brought by ours and SWA. Taking ResNet-152 as an example, the improvement brought by our cascades (1.3%; see Table 1) is higher than that brought by SWA (0.6%; see Table 2 in SWA).
> * We also think SWA and our simple ensembles or cascades are orthogonal. SWA can converge to better network weights than conventional training by using weight averaging. If we train each individual model with SWA, the accuracy of each model would increase and we believe our ensembles or cascades would also benefit from it and can achieve better accuracy.
>
>
> ### Q: Why choose max-prob as the confidence function? Is it a reasonable choice?
> **Answer:**
> * We added Figure 3 in the main text (more details in Sec D.1) to show that max-prob performs reasonably well in estimating how likely a prediction is correct.
> * We also analyze other possible confidence metrics, such as the gap between the top-2 logits or probabilities, and the (negative) entropy of the distribution in Figure 3. In addition to Figure 3, we also compare the performance of the cascade of ViT-B-224 and ViT-L-224 on ImageNet with different confidence functions Sec D.1. All these metrics are reasonable and have similar performance.
>
>
> ### Q: Is the calibration of ensembles/cascades deteriorated?
> **Answer:**
> * The calibration of ensembles/cascades is not deteriorated. Previous work [A][B] has shown that ensembles of neural networks are better calibrated than each single model. For cascades, starting from the 2nd model, we are outputting ensemble predictions, so they are also better calibrated.
>
>      [A] Simple and Scalable Predictive Uncertainty Estimation using Deep Ensembles. NeurIPS 2017.
>
>      [B] Deep Ensembles: A Loss Landscape Perspective. https://arxiv.org/abs/1912.02757.
>
> * We have empirically verified that the single models used in our experiments are actually very well-calibrated (just slightly underconfident). See Sec C.2 & Figure 5 for details. As a side note, this contradicts the common belief that deep neural networks tend to be overconfident [C].
>
>     [C] On Calibration of Modern Neural Networks. ICML 2017.

---

> ### Author Response · Authors · 2021-11-22
> **Response to Significance and Reproducibility**
>
> Thanks for the valuable comments! We are encouraged that you find our empirical study is thorough and convincing and our experimental setup is very thoughtful.
>
> ### Q: Significance compared to prior work (Kondratyuk et al., 2020; Lobacheva et al., 2020)
> **Answer:**
> * **Our empirical analysis of ensembles is much more comprehensive than prior work.** We consider more architecture families, more tasks and use large-scale benchmarks. While this extension does not have technical novelty, **it has empirical value and makes the analysis more convincing to practitioners**.
> * **We also provide a systematic analysis of model cascades with a significant amount of results, which were not discussed in their work.** They only considered model ensembles, which are not efficient at the small computation regime as shown in Figure 2. Cascades are consistently more efficient than ensembles and single models for all architecture families and at all computation regimes.
> * **Our evaluation of the inference cost is much more rigorous, which makes our results more empirically useful.** We not only consider model FLOPs but also confirm that the reduction in FLOPs can fully translate to real speedup on hardware in terms of on-device latency or throughput (Sec 5.1). This is important for real-world applications but is missing in the prior work.
>
>
> ### Q: This work would be likely very hard to reproduce. The text does not provide any training details.
> **Answer:**
> * We respectfully disagree. **We are confident that our results are very easy to reproduce** as all the models used in our experiments are **independently** trained with their **original** training procedure. We do not change the training schedule or any other hyper-parameters. We have updated our paper with more training details (see Sec B).
> * We train EfficientNet models with the official open-source implementation provided by the authors on GitHub. We train ResNet models with an open-source implementation on GitHub (https://github.com/tensorflow/tpu/tree/master/models/official/resnet).
> * **Our results hold even when one does not have access to the training code, but only to the pre-trained models.** For example, we do not train MobilNetV2 or ViT models ourselves.
> The MobileNetV2 models used in this paper are directly downloaded from its official open-source implementation. For ViT, we directly use the pre-trained checkpoints provided by the Hugging Face Team.
>
>
> ### Q: How were the ensemble models trained? Were there any specialized techniques used?
> **Answer:**
> * There are many possible ways to ensemble models, including boosting and bagging. We purposely avoid these to focus the reader on the benefits of the simplest and most known ensembling method.
> * We do not use any specialized ensemble techniques. In our experiments, we train each individual model separately with their original training scheduling and simply average their final prediction to compute the ensemble results.
>
>
> ### Q:  Deviation of the evaluation metrics
> **Answer:**
> * The deviation is quite small. All the models in our experiments are independently trained. So, the deviation of the performance of ensembles/cascades only comes from the variance of the performance of each individual model, which is very small when trained on large-scale benchmarks like ImageNet.
> * For example, we train EfficientNet-B5 architecture separately for 4 times and observe that the difference between the accuracy of the 4 B5 models is just ~0.1 %. Therefore, the ensemble of any two B5 models has very close accuracy.

---

> ### Author Response · Authors · 2021-12-01
> **Gentle reminder & Follow-up on reproducibility / the deviation of evaluation metrics**
>
> Dear Reviewer MpvP,
>
> Thanks again for your valuable review and time! Could you kindly take a look at our response and see if they have addressed your concerns? We are happy to answer any of your further questions or provide further clarifications.
>
> BTW, we have more data points to address your previous concern on reproducibility, specifically the deviation of evaluation metrics. We train 8 EfficientNet-B5 models separately and build 2-B5 ensembles/cascades using any two of these models. We observe that these different ensembles / cascades have very close performance. Here are the results:
>
> * The mean/std/max/min of the accuracy of the 8 single B5 models are:
>
> |              | mean  | std  | max   | min   |
> |--------------|-------|------|-------|-------|
> | Accuracy (%) | 83.34 | 0.04 | 83.40 | 83.29 |
>
> * The mean/std/max/min of the accuracy of the 28(=8x7/2) possible 2-B5 ensembles are:
>
> |                       | mean  | std  | max   | min   |
> |-----------------------|-------|------|-------|-------|
> | Ensemble Accuracy (%) | 84.10 | 0.05 | 84.18 | 83.97 |
>
> The FLOPs of these 2-B5 ensembles are the same (20.5B).
>
> * The mean/std/max/min of the accuracy & flops of the 56(=8x7) possible 2-B5 cascades are:
>
> |                      | mean  | std  | max   | min   |
> |----------------------|-------|------|-------|-------|
> | Cascade Accuracy (%) | 84.09 | 0.05 | 84.17 | 83.96 |
> | Cascade FLOPs (B)    | 12.62 | 0.29 | 13.35 | 12.32 |
>
> For each cascade, we tune the confidence threshold such that the cascade achieves a similar accuracy with the full ensemble.

---

> ### Author Response · Authors · 2021-12-03
> **Response to Reviewer MpvP**
>
> Thanks for your time to read our rebuttal and provide further feedback! We will continue polishing our paper and incorporate the feedback in the camera-ready version.

---

### Official Review · Reviewer_KTja · 2021-11-02

**Correctness:** 3
**Technical Novelty And Significance:** 3
**Empirical Novelty And Significance:** 3
**Recommendation:** 6
**Confidence:** 4

**Main Review:**

Using early-exit cascades to improve average efficiency, equipped with a large handful of models and means to evaluate their combinations, is a simple and practical approach.  This paper clearly demonstrates their effectiveness and profiles the degree of their effect in several scenarios, including comparisons with larger single models, NAS, and differences between small and large models.

While this paper does a good job in profiling results of this approach, I would have liked to see more on variation in the search and search space.  The exhaustive search in cascade selection appears important, but I don't see any evaluation on how important it is.  What is the performance variation among cascades in the search (particularly same-size cascades)?  What if one starts with a smaller model pool --- will results be similar or worse, and by how much, depending on the number of models trained?  Or, if one trains just 2 same-size models and uses them in a cascade with searched exit threshold, what is the expected performance and its possible variation?  How many model instances are needed in the search pool for obtaining good results with high enough probability?

The results from this paper also appear to conflict with those in Huang et al 2018 "Multi-scale dense networks for resource efficient image classification.":  On p.8 and Fig 7, that paper says "We also evaluate an ensemble of the five ResNets that uses exactly the same dynamic-evaluation procedure as MSDNets at test time ....", describing a similar cascade to this paper, and see no improvement in the average case, though the models used appear to be smaller than those investigated in this work (which this work says is a factor in effectiveness of non-cascade ensembles).  Are there any other causes for this difference?

Overall, the measurements provided here demonstrating the effectiveness of this simple and sometimes overlooked approach, are welcome evaluations.  I would have liked to see more on the importance of the search and variation between model configurations.  However, I think there is significant practical interest to see measurements that establish the effects of cascades from a candidate model pool, in the context of current classification models.



Minor questions:


sec 3 ensembles:  How many models were trained in the pool for each architecture, and how many combinations were evaluated?

sec 3 ensembles:  Were evaluations of the multiple combinations are performed on the test set, and pareto-optimal points chosen and shown as final performance for the same test set?  This would result in potentially better perf than if a separate validation set were used.  (Though in the context of the overall paper, seems a relatively minor point, since a validation set was indeed used for the cascade).


sec 7.2 semenatic segmentation:  With grid subdivisions, it's a little unclear what is the image resolution for each model, and if any models (just the first or any beyond the first) take the entire image as input to be able to incorporate larger context regions.  If 128x128 does not have enough context, for example, it seems possible the second stage of a cascade might only degrade performance compared to the initial model that used the full image.  Were there any effects like this?



**Summary Of The Paper:**

This paper investigates the effectiveness of model cascades in computation/accuracy tradeoff improvement.  A straightforward procedure is used, where all combination-permutations of a handful of models are evaluated in a cascade, and exit thresholds are determined by choosing best computation work within accuracy degradation constraint (or, best accuracy given computation constraint).  The resulting cascades perform significantly better than larger single models (more accurate or fewer flops depending on comparison).  Most significantly, the paper provides extensive evaluations on the degree of these gains for three model families (EfficientNet, Resnet, and MobileNetV2).


**Summary Of The Review:**

Overall, the measurements provided here demonstrating the effectiveness of this simple and sometimes overlooked approach, are welcome evaluations.  I would have liked to see more on the importance of the search and variation between model configurations.  However, I think there is significant practical interest to see measurements that establish the effects of cascades from a candidate model pool, in the context of current classification models.

---

> ### Author Response · Authors · 2021-11-22
> **Response - Part 2**
>
> ### Q: Results conflict with MSDNet (Huang et al., 2018)?
> **Answer:**
> * Thanks for pointing us to the interesting results in MSDNet! Our experiments on MobileNetV2 models, which are much smaller than the ResNet investigated in the MSDNet paper, also demonstrate pretty good improvement.
> * Their cascade remains the same no matter the computational budget (they always start with the smallest ResNet-10). However, in our experiments, the cascade would be different if the budget (target FLOPs) changes.
> * For example, if one wants to get B2 accuracy (small budget), then it is reasonable to start with B0. But if one wants to get B7 accuracy (large budget), starting with B0 is suboptimal. The accuracy gap between B0 and B7 is too large and B0 will not contribute any to the ensemble performance (e.g., B0+B7 ensemble does not have any accuracy gain over B7).
>
> ### Q: Were pareto-optimal ensembles selected on the test?
> **Answer:**
> * When selecting Pareto-optimal ensembles, we also use their performance on the held-out validation images, not the test images (same as we did for cascades).
>
> ### Q: Effect of grid subdivisions on semantic segmentation
> **Answer:**
> * That’s a good point! We agree it’s very possible that there is a performance drop after grid subdivisions. Due to limited time, we haven’t got a chance to qualitatively evaluate how big the performance drop could be. This might be one reason that the speedup of cascades for semantic segmentation is smaller than that for image classification.

---

> ### Author Response · Authors · 2021-11-22
> **Response - Part 1**
>
> Thanks for the insightful comments and questions! We are encouraged that you find that our work is of significant practical interest, our evaluation is extensive, and we clearly demonstrate the effectiveness of ensembles/cascades.
>
> ### Q: Why do we conduct the exhaustive search of models in the cascade?
> **Answer:**
> * We conduct the search because (1) it is cheap to do, and (2) it guarantees that an optimal configuration is found.
> * The search is cheap to do as it is conducted after all the models are independently trained. No GPU training is involved in the search. We pre-compute the predictions of each model on a held-out validation set before search. During the search, we try possible models collections by loading their predictions. We can usually find optimal model collections within a few CPU hours.
>
> ### Q: What’s the influence of the search?
> **Answer:**
> * The search helps us find much better cascades. For example, a cascade of 2 B5 models can match B7 accuracy with 13.1B FLOPs (2.8x speedup compared to B7). After search, we find that a cascade of B3+B5+B5+B5 only needs 6.9B FLOPs (5.4x speedup) to match B7 accuracy.
>
> ### Q: Cascades with same-size models
> **Answer:**
> * We compare cascades of same-size models and cascades of models of different sizes in the following tables. We use the ‘+’ notation to indicate the models used in cascades.
> * As shown in the tables, cascades with same-size models can also significantly reduce the FLOPs compared with a similarly accurate single model. But the search helps find better cascades which usually contain models of different sizes.
> * In practice, if one does not have the resources to train many models or to conduct the search, training 2 or 3 same-size models and cascading them would also be a good option.
>
> |             | Top-1 (%) | FLOPs (B) | Speedup |             | Top-1 (%) | FLOPs (B) | Speedup |
> |-------------|:---------:|:---------:|:-------:|-------------|:---------:|:---------:|:-------:|
> | B4          |    82.5   |    4.4    |         | B5          |    83.3   |    10.3   |         |
> | B3+B3+B3    |    82.6   |    2.7    |   1.6x  | B4+B4       |    83.3   |    5.1    |   2.1x  |
> | B1+B3+B4    |    82.6   |    **2.0**    |   **2.2x**  | B2+B4+B4+B4 |    83.4   |    **3.4**    |   **3.0x**  |
>
> |             | Top-1 (%) | FLOPs (B) | Speedup |             | Top-1 (%) | FLOPs (B) | Speedup |
> |-------------|:---------:|:---------:|:-------:|-------------|:---------:|:---------:|:-------:|
> | B6          |    83.7   |    19.1   |         | B7          |    84.1   |     37    |         |
> | B4+B4+B4    |    83.8   |    6.0    |   3.2x  | B5+B5       |    84.1   |    13.1   |   2.8x  |
> | B2+B4+B5+B5 |    83.7   |    **4.1**    |   **4.7x**  | B3+B5+B5+B5 |    84.2   |    **6.9**    |   **5.4x**  |
>
>
>
>
> ### Q: How large the pool of models need to be to obtain good results:
> **Answer:**
> * The EfficientNet family contains 8 architectures (B0-B7) and we train each architecture separately for 4 times. So, in total there are 32 models.
> * For ResNet, we consider 4 architectures (R50, R101, R152 & R200) and train each architecture for 2 times. So, in total there are 8 models.
> * For MobileNetV2, we directly download the pre-trained checkpoints from its official open-source implementation. We only use 5 MobileNetV2 models.
> * For ViT, we directly download the  pre-trained checkpoints provided by the Hugging Face Team. We use 4 ViT models in our experiments.
> * We see that no matter the pool of models is large or small, the cascades consistently outperform the single models as shown in Table 1 & 2.
>
> ### Q: How many combinations one needs to try:
> **Answer:**
> * The number of combinations increases exponentially as the number of models in the cascade increases. We use EfficientNet to study the influence of the number of models in cascades in Figure 6 (see appendix). We see a big gap between 2-model cascades and 3-model cascades, but increasing the number of models from 3 to 4 demonstrates a diminishing return.
> * For EfficientNet, we tried in total 4672 (= 8^4+8^3+8^2 ) possible combinations of models (we do not search over different models for the same architecture; only search the combination of architectures). But as shown in Figure 6, 3-model cascades can obtain very close performance to 4-model cascades. So one could try much fewer combinations to obtain similar results. Also, as mentioned above, evaluating one combination is cheap as no GPU training is required.
> * For ResNet and MobileNetV2, we only tried 2-model cascades due to their relatively narrow FLOPs range. Therefore, the number of possible model combinations is very small (< 20).
> * For ViT, we actually only tried 2 cascades: ViT-B-224 + ViT-L-224 and ViT-B-384 + ViT-L-384.

---

### Official Review · Reviewer_62w1 · 2021-11-03

**Correctness:** 3
**Technical Novelty And Significance:** 2
**Empirical Novelty And Significance:** 2
**Recommendation:** 6
**Confidence:** 4

**Main Review:**

## Strong points
The paper presents many experimental results and offers an interesting view on ensembling and cascades.
The paper is well written and understandable.


## Weak points
Given that efficiency is of core concern it would be good to at least discuss the concept and define what is referred to in the paper as efficient.
E.g. is it the amount of electric power used in training and inference? (Which it is not).

Given that ensembling is well-known (as correctly pointed out by the authors) and wanting to establish a new interpretation of a known concept I would expect more intuition on why the presented interpretation seems fit.

I am lacking concrete guidelines for a researcher or practitioner that go beyond existing knowledge.

While the performed experiments are rather broad in scope, the evaluation protocol is not.

Other ensembling approaches could be easily added, such as majority or variation ratio. How to set the threshold is also not explored in similar detail.

Ensembling is to me rather orthogonal to the approaches it is compared with in this paper (i.e. model scaling).


## Detailed comments
Ensembling for classification can be done in multiple different ways, see e.g. Beluch, William H., et al. "The power of ensembles for active learning in image classification." Proceedings of the IEEE Conference on Computer Vision and Pattern Recognition. 2018.

Using a confidence threshold for generating the cascade is justified with the simplicity, i would argue the approach referenced from Streeter (2018) - which is "Margin sampling"
see T. Scheffer, C. Decomain, and S. Wrobel. Active hidden Markov models for information extraction. In Proceedings of the International Conference on Advances
in Intelligent Data Analysis (CAIDA), pages 309–318. Springer-Verlag, 2001. - is also rather simple. Given the amount of experiments conducted i wonder why such a worthy comparison was skipped.

The cascades approach seems interesting, but is in my view rather is an approach how to generate an ensemble, whichs size is determined by an uncertainty measure of its first members.
Here the question arises which uncertainty measure is to be chosen, as pointed out before.

Why do you report average FLOPs to compare cascade and ensemble? How about total FLOPs over the images in the test set?

Would be good to compare the number of parameters as well.

I feel it would help the paper to strip down a bit on the amount of results included in the main text in favor of more details on why specific choices were made and conclusions taken from the results.
Some of this (also some points I raised) are hidden in the appendix, which feels suboptimal to me.


## Questions
* How was the speed-up in the tables calculated?
* How were the confidence thresholds t_i tuned?
* Sem. seg. why average confindence score? Did you try others?
* Could be interesting to plot the size of the resulting cascade over threshold

## Minor comments:

* Would be easier to read if r specifying the grid would be a different letter than r_1 and r_2 for the resolution of the self-cascade.
* Caption figure 2: typo: "cost-effectiev"

**Summary Of The Paper:**

This paper introduces ensembles as an option the reduce the amount of FLOPs while increasing or keeping the accuracy.

**Summary Of The Review:**

This paper offers a new and interesting view on the well-known ensembling of DNNs. Some details are not entirely clear to me, details see above.

---

> ### Author Response · Authors · 2021-11-22
> **Response to Other Questions**
>
> ### Q: How was the speed-up in the tables calculated?
> **Answer:**
> * In Table 1, the speedup is computed by comparing the average FLOPs of different models. We also report the speedup in terms of the on-device latency or throughput  (see Table  3&4; more results in Table 10&11) to confirm that speedup in FLOPs can fully translate to the real speedup on hardware
>
> ### Q: Semantic segmentation - why average confidence score? Did you try others?
> **Answer:**
> * In semantic segmentation, we need to predict the label for each pixel. However, the input to the network cannot be a single pixel and has to be at least an image grid. Therefore, we average the confidence score of all the pixels in the grid to indicate the overall prediction confidence of this grid. We didn’t try other choices to aggregate the confidence score of all the pixels.
>
> ### Q: Could be interesting to plot the size of the resulting cascade over threshold
> **Answer:**
> * More models will be used if we increase the confidence threshold in the cascades. We visualize several 2-model cascades in Figure 4 to understand how the thresholds influence the accuracy and computational cost of a cascade. In Figure 4, we sweep t1 from 0 to 1. When t1 = 0, the cascade reduces to the first model. When t1 = 1, the cascade becomes the same as the ensemble.
> * We also provide exit ratios of several cascades in Table 16 to better understand its behavior.
>
> ### Q: Change the notation of the grid; typos.
> **Answer:**
> * Thanks for the suggestion on the notation and pointing out the typos! We have updated them accordingly in the paper.

---

> ### Author Response · Authors · 2021-11-22
> **Response to Detailed Comments**
>
> ### Q: The power of ensembles for active learning in image classification. CVPR 2018.
> **Answer:**
> * Thanks for providing the reference. We have added a citation to this work in our paper.
> * This work compares many uncertainty estimation methods in the context of active learning, and observes that ensemble-based methods are usually the best and lead to more calibrated predictive uncertainties. This provides further incentive for practitioners to use ensembles.
>
> ### Q: Why choose maximum probability as the confidence function / uncertainty measure? Why not use “margin sampling”?
> **Answer:**
> * We added Figure 3 in the main text to compare different metrics for the confidence function ( complete details included in Sec D. 1). Possible metrics include max-probability, the gap between the top-2 logits or probabilities, and the (negative) entropy of the distribution. As shown in Figure 3,  all the metrics demonstrate reasonably good performance on measuring the confidence of a prediction, i.e., estimating how likely a prediction is correct
> * We also compare the performance of the cascade of ViT-B-224 and ViT-L-224 on ImageNet with different confidence functions in the following table (also available in Sec D.1 in the paper). For each confidence function, we set the threshold such that the cascade has a similar throughput when using different confidence functions (~409 images per second). We observe that the cascade achieves a similar accuracy with different confidence functions.
>
> |      | Top-1 (%) |
>  | :---        |    :----:   |
>  | Max Prob      | 82.3       |
>  | Logit Gap   | 82.2     |
>  | Prob Prob      | 82.3       |
>  | Entropy    | 82.1     |
>
> * According to the provided references, “margin sampling” refers to the gap between the top-2 probabilities. As shown above, it performs similarly to maximum probability.
>
> ### Q: Why do you report average FLOPs to compare cascade and ensemble? How about total FLOPs over the images in the test set?
> **Answer:**
> * By design in cascades some input examples incur more FLOPs than others. So we report the average FLOPs computed over all images in the test set. For ensembles, the FLOPs of different test images are the same.
> * The total FLOPs = average FLOPs x number of test images. For ImageNet, the total FLOPs would be average FLOPs x 50000. This does not affect the relative comparison between different models.
>
> ### Q: Would be good to compare the number of parameters as well.
> **Answer:**
> * Ensembles sometimes use fewer parameters than single models. For example, a two-B5 ensemble performs similarly to one B7 and uses fewer parameters (2 B5: 60M vs B7: 66M). But overall, ensembles are not that parameter efficient.
> * One could try using self-cascades (Table 7) if they want to reduce #parameters. For example, with self-cascades, one can obtain B7 (#params: 66M) accuracy with only one B6 model (#params: 43M).
>
> ### Q: Add more details in the main text on why specific choices were made and conclusions taken from the results.
> **Answer:**
> * We have updated the main text with more details on how we tune the confidence thresholds and the comparison of different confidence functions (see Figure 3 & Sec 4.1).

---

> ### Author Response · Authors · 2021-11-22
> **Response to Weak Points - 2**
>
> ### Q: Other ensembling approaches could be easily added, such as majority or variation ratio.
> **Answer:**
> * Averaging the predicted probabilities is a version of “soft voting”. Hard majority voting is suboptimal as it does not consider the uncertainty of a model might have in its prediction. For example, a model whose predicted class has a probability of 0.5 or 0.9 would cast the same vote in hard majority voting.
> * There are many possible ways to ensemble models, including boosting and bagging. We purposely avoid these to focus the reader on the benefits of the simplest and most known ensembling method.
>
> ### Q: How to set the threshold is also not explored in similar detail. How were the confidence thresholds t_i tuned?
> **Answer:**
> *  We determine the thresholds {t_i} on held-out validation images according to the target FLOPs or validation accuracy. In practice, we use grid search to find such thresholds.
> * Note that the thresholds are determined after all models are trained. We only need the logits of validation images to determine $\{t_i\}$, so computing the cascade performance for a specific choice of thresholds is fast. Therefore, grid search is computationally possible.
> * We have added the explanation on how to determine the thresholds in Sec 4.1 and included more complete details in Sec D.3.
>
> ### Q: Ensembling is to me rather orthogonal to the approaches it is compared with in this paper (i.e. model scaling).
> **Answer:**
> * Our point is that committee-based models are a **complementary** paradigm to achieve superior efficiency in addition to designing better architectures. Our goal is not to replace or defeat the purpose of developing better architectures.
> * When better architectures are proposed, one could build ensembles or cascades with the new architectures to obtain further improvement.

---

> ### Author Response · Authors · 2021-11-22
> **Response to Weak Points - 1**
>
> Thanks for the valuable feedback! We are encouraged that you find that our results offer an interesting view on ensembles/cascades and our paper is well written.
>
> ###  Q: What is referred to in the paper as efficient. E.g. is it the amount of electric power used in training and inference?
> **Answer:**
> * We use FLOPs (e.g., Table 1), on-device latency (Table 3&10), or on-device throughput (Table 4&11) during inference to quantify the computational cost of a model. The energy consumption (e.g., amount of electric power) can also be used to evaluate the cost of a model. We choose FLOPs, latency, or throughput in our experiments since they are easier to measure.
> * We didn’t specifically focus on the training cost. But we notice that sometimes the total training cost of an ensemble is much lower than that of an equally accurate single model (e.g., two B5 models: 96 TPU days total; one B7 model: 160 TPU days).
>
> ### Q: Given that ensembling is well-known ... I would expect more intuition on why the presented interpretation seems fit.
> **Answer:**
> * Ensembles are well-known. But the prevailing assumption is that ensembles gain accuracy by paying a larger computational cost. Our contribution is to show that this is not true and that ensembles are faster than previously assumed. We also show that converting ensembles to cascades gives a further significant speedup. **The magnitude of their efficiency compared to SOTA models is dramatic.** For example, a simple EfficientNet cascade can be **5.4x faster** than EfficientNet-B7 while being equally accurate.
> * We intentionally choose the simplest possible method to build ensembles or cascades to highlight their practical benefit. This is in sharp contrast with many previous ensembling/cascading methods, which may require a sophisticated weight generation mechanism to achieve superior efficiency(Wen et al., 2020), require the training of an early exit policy (Bolukbasi et al., 2017; Guan et al., 2018), or need to specially design a multi-scale architecture (Huang et al., 2018).
> * Our analysis empirically shows the simplicity and surprising efficiency of ensembles & cascades on several modern architecture families and large-scale benchmarks. We believe both researchers and practitioners can benefit from our analysis.
>
> ### Q: I am lacking concrete guidelines for a researcher or practitioner that go beyond existing knowledge.
> **Answer:**
> * We respectfully disagree. The existing knowledge mostly thinks that “ensembles are good in accuracy but cost more.” What we show is that this knowledge is misguided. We show that ensembles actually cost much less compared to equally accurate single models.
> * **Both the simplicity and surprising efficiency of ensembles & cascades are beyond existing knowledge**, which has been overlooked in recent literature on developing efficient neural networks. While we do not propose any new technique, such a comprehensive analysis of the efficiency of ensembles & cascades was missing from previous work.
> * Our analysis and results can immediately benefit practitioners and are also informative for researchers. The guidelines are:
>     * For neural architecture researchers - When designing a new model architecture, one should compare with an ensemble/cascade of existing architectures as a baseline. Many modern architectures would fail this test (e.g., see Table 5).
>     * For ML practitioners - When seeking better accuracy, try ensembles or cascades of your current models before chasing the newest proposed architecture.
>
> ### Q: While the performed experiments are rather broad in scope, the evaluation protocol is not.
> **Answer:**
> * We are unsure about what specific part Reviewer 62w1 wants us to improve in the evaluation protocol.
> * We have considered several vision tasks and various architecture families, **including the recent ViT architecture**. All our experiments are conducted on **large-scale benchmarks**, like ImageNet and Kinetics-600.
> * We not only compute the FLOPs, but also evaluate the **on-device latency and throughput** (Sec 5.1). We also analyze the worst-case FLOPs of cascades (Sec 5.2), as well as self-cascades, an interesting use case of cascades when only one model is available (Sec 6).
> * One interesting direction to broaden the scope of our evaluation is to extend our analysis to object detection. This is an exciting experiment we have not explored due to limited time. We leave this for future work.

---

### Public Comment · ~Andreas_Kirsch1 · 2021-11-15
**Effect of Distribution Shifts on Cascades (compared to Deep Ensembles)**

This is a great idea and a great paper overall. However, a concern after reading this paper is that it misses discussing the effects of using cascades over deep ensembles under distribution shifts. Reading the reviews, this has not been surfaced so far but has important implications for the usefulness of cascades in real-world/production environments compared to deep ensembles, for example.

Cascades are pitched as being like Deep Ensembles but as faster/more efficient essentially. Deep Ensembles are a preferred method when dealing with data uncertainties because they cope better under distribution shifts as they often appear when using ML models in real-world scenarios. Cascades lack this property though.

In particular, Deep Ensembles can expose their epistemic uncertainty by looking at the disagreement in the predictions of different ensemble members (BALD score). Points with high epistemic uncertainty are exactly points which are not fully in-distribution. At the same time, the higher the epistemic uncertainty, the higher the variance in the predictions of different ensemble members. This means that for points with high epistemic uncertainty (points that are not fully ID), single ensemble members might be overconfident or underconfident compared to the ensemble's prediction.

If the first model in the cascade happens to be overconfident for a test sample, this will trigger an early exit and will lead to worse results than computing the full ensemble each time.

This effect is hidden when using benchmark datasets as there is no distribution shift and the cascade will seem to perform about just as well as a deep ensemble.

---

Note that epistemic uncertainty is different from aleatoric uncertainty (data noise/ambiguity). The relevant equation is:

  $$\underbrace{H(\frac{1}{N} \sum_i p(y|x, \theta_i))}\_{\text{predictive entropy of ensemble}} = \underbrace{\frac{1}{N} H(\sum_i p(y|x, \theta_i))}\_{\text{average predictive entropy of each ensemble member}} + \underbrace{\frac{1}{N} \sum_i D_{KL}( p(y|x, \theta_i) || \frac{1}{N} \sum_j p(y|x, \theta_j))}_{\text{disagreement between ensemble members}}$$

where disagreement is a proxy for the epistemic uncertainty and the (average) predictive entropy tells us about the aleatoric uncertainty.

When using Bayesian models, this can be simplified to a conditional entropy and a mutual information term.

---

See also https://arxiv.org/abs/1906.02530 "Can You Trust Your Model's Uncertainty? Evaluating Predictive Uncertainty Under Dataset Shift", https://arxiv.org/abs/1703.04977 "What Uncertainties Do We Need in Bayesian Deep Learning for Computer Vision?"

---

It would be good if the draft could be updated to mention this either as a side-note/potential limitation of cascades or an opportunity for future investigations.

Thanks, \
 Andreas

---

> ### Author Response · Authors · 2021-11-22
> **Thanks for bringing this point up**
>
> Thanks for your interest in our work! This is a great point!
>
> We agree that cascades lack the good property of Deep Ensembles when only the 1st model is applied.
>
> We have empirically verified that the single models used in our experiments are actually very well-calibrated (just slightly underconfident). See Sec C.2 & Figure 5 for details. Therefore, they will not trigger early exit due to being overconfident. But we note that this analysis is done on ImageNet, i.e., in-distribution examples. Due to limited time, we haven't explored the usage of cascade on out-of-distribution data.
>
> One possible way to alleviate this issue could be instead of building cascades of single models, one could build cascades of ensembles. Then at each stage, we apply an ensemble to the test example and can get the uncertainty measurement by looking at the disagreement between ensemble members.
>
> On a side note, the fact that our single models are slightly underconfident contradicts the common belief that deep neural networks tend to be overconfident (Guo et al., 2017).
>
> “On Calibration of Modern Neural Networks. ICML 2017. Guo et al., 2017”

---

### Decision · Program_Chairs · 2022-01-20

**Decision:**

Accept (Poster)

**Comment:**

Nice paper, providing a thorough investigation of a simple idea that may be useful to a wide range of practitioners. All reviewers are positive, and the discussion has led to significant improvements in exposition and overall in the quality of the submission.